# COMFETCH: FEDERATED LEARNING OF LARGE NETWORKS ON CONSTRAINED CLIENTS VIA SKETCHING

## ABSTRACT

Federated learning (FL) is a popular paradigm for private and collaborative model training on the edge. In centralized FL, the parameters of a global architecture (such as a deep neural network) are maintained and distributed by a central server/controller to clients who transmit model updates (gradients) back to the server based on local optimization. While many efforts have focused on reducing the communication complexity of gradient transmission, the vast majority of compression-based algorithms assume that each participating client is able to download and train the current and full set of parameters, which may not be a practical assumption depending on the resource constraints of smaller clients such as mobile devices. In this work, we propose a simple yet effective novel algorithm `Comfetch`, which allows clients to train large networks using reduced representations of the global architecture via the count sketch, which reduces local computational and memory costs along with bi-directional communication complexity. We provide a nonconvex convergence guarantee and experimentally demonstrate that it is possible to learn large models, such as a deep convolutional network, through federated training on their sketched counterparts. The resulting global models exhibit competitive test accuracy over CIFAR10/100 classification when compared against un-compressed model training.

## 1 INTRODUCTION

Federated learning (FL) is an emerging setting of machine learning that has gained considerable interest within the last few years (Kairouz et al., 2019). In centralized federated learning, a set of clients, such as mobile devices, collaboratively solve a machine learning problem under the coordination of a central server without revealing any local data. This private paradigm has found use in a wide breadth of tasks such as speech prediction, document classification, computer vision, healthcare, and finance.

**Constrained Clients.** At each iteration of federated training, clients download and train a global model on privately-held local data. Model updates (in the form of weights or gradients) across all participating clients are then communicated to the central server, where they are aggregated (averaged, for example) and used to update the global model. However, an oft-neglected caveat to this procedure is that constrained clients such as mobile devices could face difficulty downloading larger models such as deep convolutional networks (Krizhevsky et al., 2009; He et al., 2016), transformers (Vaswani et al., 2017), and LSTMs (Hochreiter & Schmidhuber, 1997; Sherstinsky, 2020), which can contain prohibitive numbers of parameters. Furthermore, constrained clients may not be able to store or compute on fully-sized architectures. Consequently, deep learning towards edge-based applications face a resource challenge as state-of-the-art models continue to grow without bound (Zhao et al., 2023; Deng, 2019).

There has been an abundance of progress towards improving communication-efficiency via reduced complexity of outgoing model updates (Konečnỳ et al., 2016; Haddadpour et al., 2021; Ivkin et al., 2019; Rothchild et al., 2020; Reisizadeh et al., 2020; Horvóth et al., 2022; Safaryan et al., 2022; Khirirat et al., 2018). In all of these instances, gradients are quantized/compressed to reduce uplink complexity. However, very few works have addressed the cost of downloading and hosting large models in local client memory (downlink complexity) which is at least equally responsible for communication bottlenecks.

In this paper, we propose a novel federated learning algorithm, `Comfetch`, which allows memory-constrained clients to train a large global model by computing local gradients with respect to memory-friendly sketch-based representations of large weights. `Comfetch` parameterizes each target weight $W$ in the global architecture as a *sketch key pair* which contains a count sketch of the weight $H(W)$ and an unsketching map $\mathcal{U}(\cdot)$. Count sketches are data structures commonly used for lower-dimensional projections with desirable $\ell_2$-recovery guarantees (Charikar et al., 2002). The central server first transmits sketch key pairs of target layer weights to the client. The client then uses the key to pass each layer input $x$ first through the count sketched weight as $H(W)(x)$ and then unsketches the result using $\mathcal{U}(\cdot)$ so the output-dimensionality is retained. Since these sketched key pairs are cheaper to transmit and feed inputs through, we not only improve local memory costs but also improve communication and computational efficiency *for free*.

**Our contributions. (1)** We develop bi-directional compression FL algorithm `Comfetch` which allows memory-constrained clients to train a large network. In comparison to FL with full architecture downloads, `Comfetch` *greatly reduces memory overhead and communication costs.* Additionally, unlike other bi-directional works, *fully-sized weights are never seen in local memory.* **(2)** We experimentally demonstrate that `Comfetch` training converges to global models which are *competitive against uncompressed training* and other popular model reduction strategies such as random dropout and magnitude-based pruning while only using 10-25% of the full model size. **(3)** We provide a probabilistic *non-convex convergence guarantee* for `Comfetch`. In particular, our theory must contend with accumulated gradient approximation error resulting from successive inexact weight approximations. However, we prove under modest assumptions that these sketched architectures are guaranteed to converge to a stationary point.

## 2 PRELIMINARIES AND PROBLEM SETUP

In this section, we outline the objectives and assumptions of our federated learning setting. `Comfetch` is compatible (but not exclusively) with fully-connected networks and convolutional networks, so we also review notations common to these types of models. We end with a formal description of the Count Sketch data structure.

### 2.1 FEDERATED LEARNING SETUP

Let $\mathcal{D} = \mathcal{X} \times \mathcal{Y}$ be a global data set, where $\mathcal{X}$ and $\mathcal{Y}$ are the feature space and label space, respectively. Let $\{\mathcal{D}_i\}_{i=1}^N$ be a (possibly non-iid) collection of $N$ local client data distributions over $\mathcal{D}$. Given a loss function $\mathcal{L} : \mathcal{W} \times \mathcal{D} \to \mathbb{R}$, where $\mathcal{W}$ is a hypothesis class parameterized by weight matrices, we will solve the optimization problem,

$$\min_W f(W) = \frac{1}{N} \sum_{i=1}^N f_i(W), \tag{1}$$

where $W$ describes the set of model parameters and $f_i(W) = \mathbb{E}_{z \sim \mathcal{D}_i} \ell(W; \xi)$ is client $i$'s loss function $\ell$ (we assume homogeneous loss type) and local data distribution $\mathcal{D}_i$. The central server and clients will collaboratively solve this optimization problem in an iterative manner, so we let $W_t$ represent the global model weights at time $t$. At the beginning of each round, the server selects $N$ clients uniformly at random from a large cluster to participate in training. Each client $c_i$ for $1 \leq i \leq N$ downloads $W_t$ and minimizes $f_i$ using a preferred optimizer (SGD, Adam, etc.) and sends their locally-updated model parameters (or gradients) to the central server. The server aggregates all the local models to update the global weights. A typical scheme for aggregation is averaging over models/gradients (as implied in equation 1), which is referred to as FedAvg (McMahan et al., 2017).

### 2.2 NETWORK ARCHITECTURES

To facilitate later descriptions of how to parametrize layer weights via the count sketch, we review the forward pass of two popular architectures: fully-connected networks and convolutional ResNets. Our description follows the notations of (Du et al., 2019).

*Multilayer fully-connected networks:* Let $\{W_t^\ell\}_{\ell=1}^L$ represent the weights of our layers at time $t$, where $L$ is the depth of the network. Let $x \in \mathbb{R}^d$ be the input. We define the network prediction

recursively. Let $\sigma : \mathbb{R}^d \to \mathbb{R}^d$ represent a nonlinear activation function, and denote $x^0 := x$. We have that $x^\ell = \sigma(W_t^\ell x^{\ell-1})$ for $1 \le \ell \le L - 1$ and the final prediction is $\hat{y} = \mathbf{a}^\top x^L$, where $\mathbf{a} \in \mathbb{R}^d$ is the output layer. Here, we have omitted bias and regularization terms. The output $\hat{y}$ is fed into $\mathcal{L}$ where it is compared against the true label.

*Convolutional ResNet:* We will now describe the output of a canonical ResNet architecture. We will intentionally avoid using any convolutional operators ($*$) since we will be sketching along the input or output channel mode. Let $x^0 \in \mathbb{R}^{s_0 \times p}$, where $s_0$ is the number of input channels and $p$ is the number of pixels. We denote $s^\ell = m$ as the number of channels and $p$ as the number of pixels for all $\ell \in [L]$. For $x^{\ell-1} \in \mathbb{R}^{s_{\ell-1} \times p}$, we use an operator $\phi_\ell(\cdot)$ to divide $x^{\ell-1}$ into a stack of $p$ patches. Each patch will have size $qs_{\ell-1}$ which implies that $\phi_\ell(x^{\ell-1}) \in \mathbb{R}^{qs_{\ell-1} \times p}$. Let $W_t^\ell \in \mathbb{R}^{s_\ell \times qs_{\ell-1}}$. Similar to the fully-connected case, we define the layers recursively:

$$x^1 = \sqrt{\frac{c_\sigma}{m}}\sigma\Big(W_t^1 \phi_0(x^0)\Big),$$
$$x^\ell = x^{\ell-1} + \frac{c_{res}}{L\sqrt{m}}\sigma\Big(W_t^\ell \phi_\ell(x^{\ell-1})\Big), \qquad 2 \le \ell \le L,$$

$0 < c_{res} < 1$. The output is $\hat{y} = \langle W_t^L, x^L \rangle$, where $W_t^L \in \mathbb{R}^{m \times p}$ and $\langle , \rangle$ is the Frobenius product.

## 2.3 COUNT SKETCH

The crux of `Comfetch` is using the count sketch to compress layer weights. Briefly, the count sketch data structure contains a collection of $k$ pairwise-independent hashing maps $h_i : [d] \to [c]$ for $i \in [k]$, each of which is paired with a sign map $s_i : [d] \to \{\pm 1\}$, for $i \in [k]$. Each hash function/sign map pair is used to project a $d$-dimensional vector $x$ into a smaller $c$-dimensional space, which we refer to as a sketch. Through an "unsketch" procedure (detailed in Appendix A), we use the $k$ sketches to create a $d$-dimensional approximation $\hat{x}$ of $x$. To develop a sketch of a weight matrix, we apply a count sketch to each row of the weight matrix. Our method is generalizable to tensorial weights, but these require higher-order count sketches (HCS); we refer the reader to Appendix A for further treatment of HCS.

## 3 RELATED WORK

### 3.1 NETWORK COMPRESSION

A popular approach to network compression involves taking low-rank factorizations of the weight tensors (Oseledets, 2011; Denil et al., 2013; Tai et al., 2015; Novikov et al., 2015). Oftentimes, this will require learning the factors, which increase training overhead (Oseledets, 2011; Denil et al., 2013) or increases the depth of the network (Tai et al., 2015), and computing exact tensor factorizations is known to be NP-hard (Gillis & Glineur, 2011). `Comfetch` avoids these issues by relying on simple linear transformations of the original weights. (Kasiviswanathan et al., 2017) replaces fully-connected and convolutional layers in a non-federated setting via sign sketches (Hadamard matrices). Their compressed CNNs perform worse than our federated models on CIFAR-10 (Krizhevsky et al., 2009) classification, cost more memory than our `Comfetch` to store, and do not have convergence guarantees. Knowledge distillation (Hinton et al., 2015; Ba & Caruana, 2013), quantization (Jacob et al., 2018), binarization (Hubara et al., 2016), Huffman coding (Han et al., 2015), and other similar techniques seek to gradually reduce the number of the parameters during training or compress the model post-training. In our scenario, the memory-constrained clients can never store the original architecture due to its size, and in most centralized FL settings, the server does not have access to the local client data Kairouz et al. (2019), therefore rendering these techniques inapplicable. Hence, we opt for the popular count sketch compression scheme, which is data-oblivious.

### 3.2 COMMUNICATION-EFFICIENT FEDERATED LEARNING

A practical assumption of centralized federated learning is that clients will be physically removed from the central server and communicate over unreliable wireless channels (Kairouz et al., 2019; Yang et al., 2018; Amiri & Gündüz, 2020). Therefore, there has been significant interest in reducing the size of data communicated between the server and federated agents (Konečný et al., 2016).

(Ivkin et al., 2019) suggest taking count sketches of the local gradients to reduce client update costs to great effect, but their algorithm `Sketched SGD` requires an extra round of communication with the server, which although appropriate for their distributed single-machine setting, would fail in the general federated setting due to a lack of persistent clients. (Rothchild et al., 2020) successfully eliminates the extra round with their `FetchSGD` by taking several independent sketches of the gradients coupled with error feedback (Karimireddy et al., 2019) in the server update phase. `FetchSGD` only extracts the Top-$k$ components of the gradients to mitigate count sketch recovery error, similar to the `MISSION` algorithm (Aghazadeh et al., 2018). Low-precision quantization of gradients has also been proposed (Alistarh et al., 2017; Reisizadeh et al., 2020) to great effect.

While the aforementioned methods successfully decrease upload communication costs, they do not address downlink complexity, which could be a bottleneck for memory-constrained clients such as mobile devices. (Niu et al., 2020) and (Diao et al., 2021) propose distributing subnetworks of the global models to clients based on computational and memory constraints, but we are interested in implicitly preserving the full-dimensionality of the model to maximize performance. (Shah & Lau, 2021) sparsifies global models at the server, which once again, reduces the overall capacity of the model, and is better suited for pruning; pre-trained models. *Bi-directional compression works are still uncommon*: to the best of our knowledge, (Dorfman et al., 2023), is the only other work to consider compression of model weights while retaining dimensionality, but convergence guarantees are only provided under aggressive assumptions of lossless decompression and experiments are only performed on relatively small models such as ResNet-9. Other bi-directional works transmit compressed gradients (Philippenko & Dieuleveut, 2020; Tang et al., 2019; Gruntkowska et al., 2023; Zheng et al., 2019) and require restoration inside local memory which is equivalent to storing a fully-sized architecture, which `Comfetch` avoids.

## 4 COMFETCH

We propose Algorithm 1, `Comfetch`, to minimize the aggregated loss function (equation 1) of a large network on memory-constrained devices. Algorithm 1 uses only a single sketch, but it is possible to utilize multiple sketches of weights. We assume a single count sketch is used for all weight compressions for the remainder of this section.

We use a sketch-unsketch paradigm $W_t^\ell \to \mathcal{U}(H(W_t^\ell)) = H^{\ell\top} H^\ell W_t^\ell$, where $H^\ell \in \mathbb{R}^{d \times c}$ is a randomly drawn count sketch matrix to approximate each weight $W_t^\ell \in \mathbb{R}^{d \times d}$ at iteration $t$ and layer $\ell$ with $c << d$. We require the client to store only a hash-function $h^\ell \in \mathbb{R}^d$ associated with $H^\ell$ and the sketched weight $H^\ell W_t^\ell$, thus reducing the memory footprint of storing each layer from $\mathcal{O}(d^2)$ to $\mathcal{O}((c+1)d)$. (As an abuse of notation, we use $h^\ell$ and $H^\ell$ interchangeably, since one may infer $H^\ell$ from $h^\ell$, the latter of which we will transmit in implementation since it is cheaper to do so.) We design a mechanism for the central controller to aggregate model updates from the clients and backpropagate the sketched model parameters. The algorithm follows the typical structure of a federated learning algorithm: *model transmission and download*, *client update*, and *model update*. This process is repeated over $T$ iterations. We will describe each phase in detail for a fixed weight. For simplicity of notations, we assume that $W_t^\ell \in \mathbb{R}^{d \times d}$.

### 4.1 MODEL TRANSMISSION AND DOWNLOAD

At iteration $t$, the central server first prepares the global model for the transmission by sketching down all the current weights $\{W_t^\ell\}_{\ell=1}^L$, via Count Sketch matrices $H^\ell \in \mathbb{R}^{c \times d}$, where $c << d$ is referred to as the *sketching length* or *sketch dimension*. We assume that our layers are either convolutional or fully-connected as described in Section 2. For each weight $W_t^\ell$, the server randomly draws a count sketch matrix $H^\ell$. The server transmits $\{(H^{\ell\top}, H^\ell W_t^\ell)\}_{\ell=1}^L$ to $N$ clients selected uniformly at random from a large cluster, who then download the sketched parameters.

**Cost Complexity.** Note that any $H^\ell$ bijectively corresponds to a hash function $h^\ell : d \to c$ which is representable as a length $d$ vector, so it is cheaper to store/transmit $h^\ell$. Hence, in practice, the central server will transmit $\{(h^\ell, H_i^\ell W_t^\ell)\}_{\ell=1}^L$, for a total local memory and transmission cost of $\mathcal{O}((c+1)d)$, which is far less than the usual $O(d^2)$ cost of transmission and storage.

---

**Algorithm 1** `Comfetch` (Single Sketch)

---

**Require:** initial weights $\{W_0^\ell\}_{\ell=1}^L$, learning rate $\eta$, number of iterations $T$, momentum parameter $\rho$, batch size $M$ of data, batch size $N$ of clients

    Init momentum term $\{u_0^\ell = 0\}_{\ell=1}^L$

2: Init error accumulation term $e_0 = 0$

    **for** $t = 1, 2, \ldots, T$ **do**

4:        Init sketch key pairs $\{H^{\ell^T}, H^\ell W_t^\ell\}_{\ell=1}^L$

        Uniformly select at random $N$ clients $c_1, c_2, \ldots, c_N$

6:        **loop** $\{$in parallel on clients $\{c_i\}_{i=1}^N\}$

        **for** $\ell = 1, 2, \ldots, L$ **do**

8:            Download weight sketches $\{H^{\ell^T}, H^\ell W_t^\ell\}_{\ell=1}^L$

            Compute grads $g_i^\ell = \nabla_{H^\ell w_t^\ell} \mathcal{L}(\hat{W}_t^\ell, z \sim \mathcal{D}_i)$          $\triangleright \hat{W}_t^\ell = H^{\ell^\top} H^\ell W_t^\ell$

10:      **end for**

        Send $\{g_i^\ell\}_{\ell=1}^L$ to Central Server

12:      **end loop**

        **for** $\ell = 1, 2, \ldots, L$ **do**

14:          Aggregate restored gradients: $g^\ell = \frac{1}{N} \sum_{i=1}^N g_i^\ell$

            Update momentum: $u_t^\ell = \rho u_{t-1}^\ell + g^\ell$

16:          Update error feedback: $e_t^\ell = \eta u_t^\ell + e_t^\ell$

            Approximate gradient: $\Delta_t = \text{Top-}k(e_t^\ell)$

18:          Error accumulation: $e_{t+1}^\ell = e_t^\ell - \Delta_t$

            Update weight: $W_{t+1}^\ell = W_t^\ell - \Delta_t$

20:      **end for**

    **end for** **return** $\{W_T^\ell\}_{\ell=1}^L$

---

## 4.2 CLIENT UPDATE

The client $C_i$ will now conduct a single round of training on the sketched network parameters using local data. In practice, the client distributions $\mathcal{D}_i$ will be finite and small (Kairouz et al., 2019), so we assume that the client is taking the full gradient with respect to the weights, but the algorithm generalizes to stochastic gradients as well.

**Figure 1:** The forward pass of a fully-connected layer.

**Forward pass.** Assume there are $L$ weights. Following the notations described in Section 2, the forward pass of a fully-connected (FC) layer is $x^\ell = \sigma(H^{\ell^\top}(H^\ell W_t^\ell x^{\ell-1}))$, for $1 \leq \ell \leq L-1$ and the final output is $\hat{y} = \mathbf{a}^\top x^L$. Similarly, for a convolutional ResNet, we have that

$$x^1 = \sqrt{\frac{c_\sigma}{m}} \sigma\left(H^{1^\top}(H^1 W^1 \phi_1(x^0))\right), \tag{2}$$

$$x^\ell = x^{\ell-1} + \frac{c_{res}}{L\sqrt{m}} \sigma\left(H^{\ell^\top}(H^\ell W^\ell \phi_\ell(x^{\ell-1}))\right), \tag{3}$$

for $2 \leq \ell \leq L$ and the final output is $\hat{y} = \langle W^L, x^L \rangle$, where $W^L \in \mathbb{R}^{m \times p}$.

**Cost Complexity.** Each sketched weight costs the client $\mathcal{O}(cd)$ to locally store which is an improvement over storing the original weight which costs $\mathcal{O}(d^2)$.

*Remark 1:* The client never sees or directly compute a $d \times d$ weight matrix at any stage. As emphasized by parenthetical grouping, for example in the case of a fully-connected layer, we compute $H^\ell W_t^\ell x^{\ell-1}$, followed by multiplication on the left by $H^{\ell^\top}$.

*Remark 2:* In the convolutional case, where the kernel can be interpreted as a tensor of filter weights, $H^\ell W^\ell \phi_\ell(x^{\ell-1})$ can be regarded as a higher-order count sketch (HCS) of $W^\ell \phi_\ell(x^{\ell-1})$ (Appendix A.2, or we may matricize the kernel and apply the count sketch in a stacked manner.

## 4.3 BACKWARDS PASS AND UPLINK

To compute the gradient, we must first clearly define the weights of the client networks. We have $H^{\ell\top} H^\ell W_t^\ell x^{\ell-1}$ and $H^\ell W^\ell \phi_\ell(x^{\ell-1})$ in the fully-connected and convolutional layers, respectively. Therefore, we can represent each sketched weight as $R^\ell W_t^\ell$ where $R^\ell = H^{\ell\top} H^\top W_t^\ell$, so our sketches are simply dimension-preserving linear transformations of the original weights. For further notational convenience, we denote $\hat{W}_t^\ell \triangleq R^\ell W_t^\ell$. Now that we have defined the weights of our client models, we may now take gradients. The server will want to receive $\frac{\partial \mathcal{L}(\hat{W}_t^\ell, z)}{\partial W_t^\ell}$ as an approximation of $\frac{\partial \mathcal{L}(W_t^\ell, z)}{\partial W_t^\ell}$, but the client will not want to store or compute $\frac{\partial \mathcal{L}(\hat{W}_t^\ell, z)}{\partial W_t^\ell}$, since it is of size $d \times d$. Instead, the client will transmit a $\mathcal{O}(c \times d)$ packet of data which will allow the server to compute $\frac{\partial \mathcal{L}(\hat{W}_t^\ell, z)}{\partial W_t^\ell}$. Using the chain rule we have that:

$$\frac{\partial \mathcal{L}(\hat{W}_t^\ell, z)}{\partial W_t^\ell} = \frac{\partial \mathcal{L}(\hat{W}_t^\ell, z)}{\partial H^\ell W_t^\ell} \frac{\partial H^\ell W_t^\ell}{\partial W_t^\ell}. \tag{4}$$

Since the server has knowledge of $\frac{\partial H^\ell W_t^\ell}{\partial W_t^\ell} = H^\ell$, the client only needs to upload $g_i^\ell \triangleq \nabla_{H^\ell W_t^\ell} \mathcal{L}(\hat{W}_t^\ell, z) \in \mathbb{R}^{c \times d}$. One might ask: *how does $\nabla_{H^\ell W_t^\ell} \mathcal{L}(\hat{W}_t^\ell, z)$ relate to $\nabla \mathcal{L}(\hat{W}_t^\ell, z)$?* Using the chain rule again (and dropping $z$ for notational convenience), we have that

$$\nabla_{W_t^\ell} \mathcal{L}(\hat{W}_t^\ell, z) = \nabla \mathcal{L}(\hat{W}_t^\ell) \nabla_{W_t^\ell} \hat{W}_t^\ell = \nabla \mathcal{L}(\hat{W}_t^\ell) R^\ell, \tag{5}$$

which indicates that $\nabla_{W_t^\ell} \mathcal{L}(\hat{W}_t^\ell) R^\ell = \nabla \mathcal{L}(\hat{W}_t^\ell)$, i.e., the gradients the client is submitting to the server are count sketch approximations of the true gradient of our sketched network. Thus, we are performing uplink compression of our gradients using count sketches as our compression operator.

**Cost Complexity.** Each $g_t^\ell$ costs $\mathcal{O}(cd)$ to store and transmit, which is a strong improvement over the usual uplink complexity of $\mathcal{O}(d^2)$ and even cheaper than the sketched storage cost of $\mathcal{O}((c+1)d)$. *Remark.* The client will never directly compute $g_i^\ell \in \mathbb{R}^{d \times d}$. The purpose of equation equation 4 is to illustrate the gradient calculation, but we first take care to show this derivative is well-defined. Assuming $\mathcal{L}$ is differentiable with respect to any weight, we only need to prove that $\frac{\partial R^\ell W}{\partial H^\ell W}$ is computable, which is indeed the case due to the structure $H^\ell$. In particular, for any valid count sketch matrix $H\mathbb{R}^{c \times d}$ and vector $x \in \mathbb{R}^d$, if $x_i$ is bucketed by hash function $h_j$, we have that $[H^\top H x]_i = H_{ji} \cdot [H x]_j$, therefore, the partial derivative $\frac{\partial R^\ell W}{\partial H^\ell W}$ is well-defined. However, in practice, the client will use an autograd-like library.

## 4.4 MODEL UPDATE

The Central Server aggregates the $\{g_i^\ell\}_{\ell=1}^L$ across all clients $c_i$ for $i \in [N]$ and computes a decompressed average over the gradients: $g^\ell = \frac{1}{N} H^{\ell\top} \sum_{i=1}^N g_i^{\ell\top}$.

The remainder of the model update is an SGD (or Adam)-like procedure that follows the error-feedback and momentum scheme similar to other compression-correcting literature (Rothchild et al., 2020; Ivkin et al., 2019). The error-feedback term $e_t$ allows for the correction of error associated with our gradient approximations $g^\ell$. Specifically, we are correcting the error associated with using $\nabla_{W_t^\ell} f(R^\ell W_t^\ell)$ as an approximation of $\nabla f(W_t^\ell)$. Once we form the full error term $e_t$, we take the Top-$k$ components (in absolute magnitude) of it, which we expect to be relatively undiluted by the approximation error, to form $\Delta_t$. We have that $\Delta_t$ is our error-corrected gradient approximation with a momentum term already baked into it. (The momentum term $u_t$ is common to SGD-variants in the non-federated setting, the benefits of which are discussed by Sutskever et al. (Hinton et al., 2015).) This $\Delta_t$ will help us mimic stochastic gradient descent, as shown in Line 18 of Algorithm 1.

## 5 CONVERGENCE GUARANTEE

In this section, we provide a non-convex convergence result for `Comfetch` For all results, $|| \cdot ||$ refers to the $\ell_2$ norm. We begin by outlining our assumptions. Without loss of generality, we will denote our weights as $w \in \mathbb{R}^d$ (through vectorization, for example).

**Assumption 1** ($L$-Smooth). *The objective function $f(W)$ in equation 1 is L-smooth. That is, for all $x, y \in \mathbb{R}^d$ we have that,*

$$||\nabla f(x) - \nabla f(y)|| \leq L||x - y||. \tag{6}$$

**Assumption 2** (Unbiased and Bounded). *All stochastic gradients $g$ of $f(w)$ are unbiased and bounded,*

$$\mathbb{E}g = \nabla f(w) \quad and \quad \mathbb{E}||g||^2 \leq G^2. \tag{7}$$

Assumptions 1-2 are standard to convergence proofs of SGD-like algorithms, including those of a federated nature (Karimireddy et al., 2019; Nemirovski et al., 2009; Ivkin et al., 2019; Shalev-Shwartz et al., 2011; Rothchild et al., 2020).

**Assumption 3** (Heavy Hitters). *In the notation of Algorithm 1, let $\{W_t^\ell\}_{t=1}^T$ be the sequence of model weights of the $\ell^{th}$ layer generated by `Comfetch`. There exists a constant $\epsilon$ such that for all $t \in [T]$, the approximated gradient of our sketch network with momentum $z_t^\ell := \eta(\rho u_{t-1}^\ell + g_{t-1}^\ell) + e_{t-1}^\ell$ contains at least one coordinate $i$ such that $(z_t)_i^2 \geq \epsilon ||z_t||^2$. Furthermore, there exists a constant $c$ such that for any given weight $W_t^\ell$, there exists a coordinate $j$ with $(W_t^\ell)_j^2 \geq (1/c)||W_t^\ell||^2$. These coordinates are referred to as heavy hitters (Alon et al., 1999).*

Assumption 3 is a variant on a common heavy-hitter assumption suggested in the convergence theorem of `FetchSGD` to ensure successful error-feedback (Rothchild et al., 2020). Heavy hitters are also required in the convergence analysis of `Sketched-SGD` (Ivkin et al., 2019). In this version, we are requiring $\nabla_w \mathcal{L}(H^\top H w)$ to contain a heavy hitter.

**Theorem 1.** *Let $w_0 \in \mathbb{R}^d$ denote an initialized model weight and consider a sketch of size $\mathcal{O}\left(\frac{1}{\epsilon^2} \log \frac{d}{\delta}\right)$ where sketch dimension $c = 1/\epsilon^2$. Define $\tilde{f}(x) = f(\mathcal{U}_H(H(x))$. Under Assumptions 1-3 and with step size $\gamma = \frac{c(1-\rho)}{2Ld\sqrt{T}}$, we have that Comfetch returns $\{w_t^\ell\}_{i=1}^T$ such that*

$$\min_{t=1\cdots T} ||\nabla \tilde{f}(w_t)||^2 = \mathcal{O}\left(\frac{4Ld(f(w_0) - f^*) + G^2}{c\sqrt{T}} + \frac{2d^2(1+\epsilon)^2 G^2}{c^2(1-\epsilon)^2 \epsilon^2 T}\right), \tag{8}$$

*with probability $1 - \delta$ over the sketching randomness.*

Our analysis critically relies on the fact that if our original network is $L$-smooth, the sketched architecture is $\frac{Ld}{c}$-smooth. We defer the proof to Appendix B

## 6 EXPERIMENTS

In this section, we investigate the performance of `Comfetch` models.

**Vision Task.** We perform image classification tasks over CIFAR-10 and CIFAR-100 (Krizhevsky et al., 2009) using a ResNet-18 architecture, which contains roughly 11M parameters. Both CIFAR-10/100 are benchmark computer vision datasets containing with 60K 32×32 color images labeled with 10 or 100 possible labels, respectively. Vision experiments were run on a computing cluster using mixtures of NVIDIA Tesla T4 and RTX A4000 GPUs each equipped with 16 GB RAM. We used the PyTorch library for training our models and MPI for distributed averaging.

**Training Setup.** The number of federated clients is always either 4 or 10, which we specify within the captions. At each round, all clients download the current model and each weight $W$ is passed through a count sketch recovery operation $H^\top H W$, where $H$ is a count sketch matrix of size $d_2 \times cr \cdot d_1$, where $d_2$ is the output dimensionality, $cr$ is the compression rate, and $d_1$ is the input dimensionality. This simulates a layer input passing through a sketch key pair (see Figure 1).

Clients will locally train their model for $E = 1$ epoch with batch size $B = 128$ before averaging and the clients use an Adam optimizer with learning rate $lr = 0.001, \beta_1 = 0.9, \beta_2 = 0.999, \epsilon = 1e - 8$. After completing the training epoch, the local models are averaged to update the global model.

The sketch used is homogeneous across all clients and we do not use error feedback (which has been observed to have an insignificant effect on performance and only needed for theoretical analysis).

**Data Splits.** For iid training, the training dataset is shuffled and then each client is handed $|\mathcal{D}|/N$ samples selected uniformly at random, where $|\mathcal{D}|$ is the total dataset size and $N = 4$ or $N = 10$. For non-iid training, the size of local training sets are the same size, but samples are selected in a label-skewed manner according to a Dirichlet allocation.

## 6.1 CIFAR10/100 EXPERIMENTS

*In general,* `Comfetch` *models are competitive against uncompressed FedAvg training in CI-FAR10/100 training.* CIFAR-10 model performance is presented in Table 1 and and Figures 2 and 3. For iid settings, 50% sketched compression results in a test accuracy drop off of $< 2\%$, while 75% compression results in decreased accuracy of $< 4\%$. We begin to notice a significant decline at 90% compression, which appears to be a general threshold across all experiments. Non-iid training appears to be slightly more challenging, but nonetheless, our `Comfetch` models mimic the performance of un-sketched models with minimal dropoff for compression rates up to 75%. CIFAR-100

| Method | Compression Rate | IID Test Accuracy (%) | Non-IID Test Accuracy (%) |
|---|---|---|---|
| Comfetch | 90% | 76.53 | 75.29 |
| Comfetch | 75% | 82.85 | 81.89 |
| Comfetch | 50% | 84.42 | 83.59 |
| No Compression | 0% | 86.37 | 84.99 |

**Table 1: IID CIFAR-10 Top accuracy.** Average top test accuracy over three runs for 10 client `Comfetch` classifying CIFAR-10 Krizhevsky et al. (2009) using ResNet-18. We analyze how `Comfetch` performs over a range of compression rates as well as IID and non-IID ($\alpha = 1$) dataset splits.

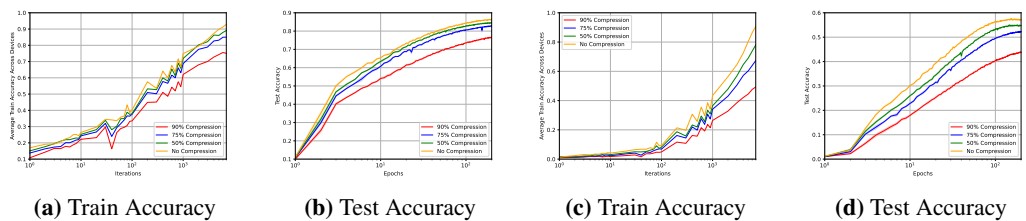

| **(a)** Train Accuracy | **(b)** Test Accuracy | **(c)** Train Accuracy | **(d)** Test Accuracy |
|---|---|---|---|

**Figure 2: IID CIFAR-10/100 Training/Test Curves.** Test accuracy convergence of `Comfetch` under varying compression rates. **(a)-(b)** corresponds to CIFAR-10 Krizhevsky et al. (2009) image classification, while **(c)-(d)** correspond to CIFAR-100 image classification. In these experiments, only a single sketch is used and the datasets are IID. Similar accuracy with different `Comfetch` compression rates suggests that our method retains the expressive power of non-sketched models, while simultaneously reducing their storage size.

training, displayed in Table 2 and the bottom series of Figures 2 and 3, is far more challenging. However, our models still display competitive performance against non-sketched training for models with up to 75%, regardless of iid or non-iid training. We similarly observed a noticeable decline at 90%

| Method | Compression Rate | IID Test Accuracy (%) | Non-IID Test Accuracy (%) |
|---|---|---|---|
| Comfetch | 90% | 43.80 | 42.52 |
| Comfetch | 75% | 52.25 | 50.21 |
| Comfetch | 50% | 54.76 | 52.89 |
| No Compression | 0% | 57.16 | 55.21 |

**Table 2: Non-IID CIFAR-100 Top Accuracy.** Average test accuracy over three random runs for 10 clients classifying CIFAR-100 Krizhevsky et al. (2009) using ResNet18. We analyze how `Comfetch` performs over a range of compression rates as well as IID and non-IID ($\alpha = 1$) dataset splits.

compression, which we suspect is due to the sketch recovery becoming too lossy at such an extreme.

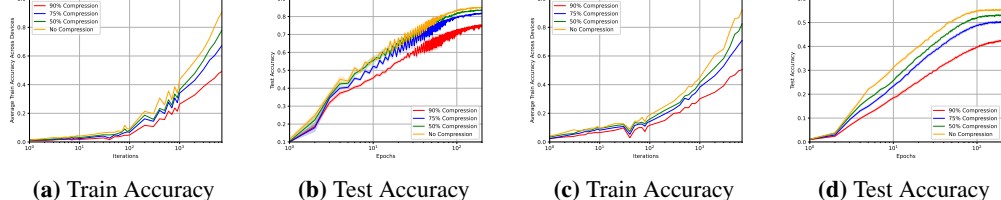

**(a)** Train Accuracy      **(b)** Test Accuracy      **(c)** Train Accuracy      **(d)** Test Accuracy

**Figure 3: Non-IID CIFAR-10/100 Training/Test Curves.** Test accuracy convergence of `Comfetch` under varying compression rates in non-IID settings. **(a)-(b)** corresponds to CIFAR-10 image classification, while **(c)-(d)** corresponds to CIFAR-100 image classification. Datasets are non-IID with a Dirichlet split using $\alpha = 1$. Furthermore, only a single sketch is used. Similar accuracy with different `Comfetch` compression rates suggests that our method retains the expressive power of the model while reducing the parameter counts even in the non-IID domain.

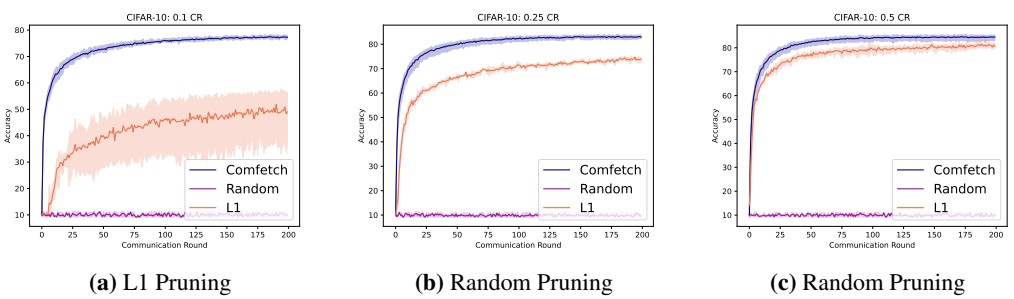

**(a)** L1 Pruning        **(b)** Random Pruning        **(c)** Random Pruning

**Figure 4: Pruned Models.** 4 clients are given iid distributions of CIFAR-10 (12500 samples each) split across classes. Client model weights are then pruned randomly or based on L1 metrics at compressions of 0.1, 0.25 and 0.5.

## 6.2 FEDAVG PRUNING

For multi-client federated averaging, we investigate baseline model compressions in Figure 4 where a non-sketched ResNet-18 is pruned according to $\ell_1$-magnitude or random dropout prior to training to simulate compression. There are 4 clients in this setting, splitting CIFAR-10 in an iid manner. While such strategies have been shown to perform well for pre-trained models, we demonstrate in Figure 4, that such naive model compression is less performant than Comfetch. *No matter the pruning strategy or degree of compression, FedAvg models compressed in this manner underperform sketch-based weight compression.* Random dropout, in particular, prior to training is especially destructive. We observe that at 90% compression with random weight pruning, the models essentially classify close to random. Magnitude-based pruning appears to be far superior to random pruning.

## 7 CONCLUSION

In this work, we present a federated learning algorithm `Comfetch` for training large networks on memory-constrained clients. In our scheme, the central server parameterizes the global model via sketch key pairs of the weight matrices, significantly reducing storage costs. These sketched architectures greatly reduce bi-directional communication, local memory, and computational costs while retaining high performance.

The limitations of `Comfetch` motivate future directions. We note that the theory developed in Section 5 does not predict the success of single-sketch `Comfetch` which we observe to be very effective in our experiments. Structurally-aware sketches (Zhang et al., 2020; Chen & Zhang, 2016) or oblivious sketches (Ahle et al., 2020) may provide insight into one-sketch guarantees.

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

APPENDIX

# A SKETCHES

In this section, we review sketching algorithms and data structures which are the most relevant to the implementation of `Comfetch`.

## A.1 COUNT SKETCH

Sketching algorithms seek to compress high-dimensional data structures into lower-dimensional spaces via entry hashing. Sketching was initially proposed by Muthukrishnan et al. (Cormode & Muthukrishnan, 2005) as a solution to estimating the frequency of items in a stream Cormode & Muthukrishnan (2005). Alon et al. Alon et al. (1999) then proposed the eponymous AMS sketch to formalize this setting along with $\ell_2$-guarantees of the decompression of hashed data. The *Count Sketch* is the successor of the AMS sketch, first proposed by Charikar et al. (Charikar et al., 2002) and outlined in Algorithm 2. The Count Sketch is desired in settings where one is interested in estimating the heavy hitters of a high-dimensional vector, which are the entries with large absolute magnitude relative to other entries.

The Count Sketch data structure contains a collection of $k$ pairwise-independent hashing maps $h_i : [d] \to [c]$ for $i \in [k]$, each of which is paired with a sign map $s_i : [d] \to \{\pm 1\}$, for $i \in [k]$. Each hash function/sign map pair is used to project a $d$-dimensional vector $x$ into $c$-dimensional space, which we refer to as a sketch. Furthermore, we refer to $c$ as the sketching length and $k$ as the number of the sketches. Each of the $k$ sketches is representable as a linear transformation $H_i x \in \mathbb{R}^c$ for $i \in [k]$, where $H_i \in \mathbb{R}^{c \times d}$ is referred to as a Count Sketch matrix. To "unsketch" or restore a sketch back to $d$-dimensional space, we simply multiply by the transpose of $H_i$, i.e., $\hat{x}_i = H_i^\top H_i x \in \mathbb{R}^d$, where we use the hat notation to specifically indicate that $\hat{x}_i$ is merely an approximation of $x$. The final approximation $\hat{x}$ of $x$ is determined coordinate-wise as $(\hat{x})_j = \underset{1 \leq i \leq k}{\mathrm{median}}\{\hat{x}_i\}_{i=1}^k$ for $j \in [d]$.

**Count sketch matrices** As detailed in the previous section, each sketch has a matrix representation which we will now make explicit. Let $h : [d] \to [c]$ and $s : [d] \to \{\pm 1\}$ be a hash function/sign pair. The associated count sketch matrix $H$ is representable as the following $c \times d$ matrix transformation: fixing $i \in [c]$ and $j \in [d]$, we have that $(H)_{ij} = s(i)$ if $h(j) = i$ for $j \in [d]$ and 0 otherwise.

---

**Algorithm 2** `Count Sketch` Charikar et al. (2002)

---

**Require:** vector $x \in \mathbb{R}^d$, number of hash functions $k$, sketch length $c$
    **procedure** INIT($c, k$)
2:      Init sign hashes $\{s_j\}_{j=1}^k$ and hash functions $\{h_j\}_{j=1}^k$        ▷ Must be 2-wise independent
    **end procedure**
4:      Init $k \times c$ table of counters $S$
    **procedure** SKETCH($i, x_i$)
6:    **for** $j = 1, \ldots, k$ **do**
        $S[j, h_j(i)] += s_j(i) x_i$
8:    **end for**
    **end procedure**
10: **procedure** UNSKETCH($k$)
      Init length $k$ array estimates
12:    **for** $j = 1, \ldots, k$ **do**
        estimates$[k] = s_j(i) S[j, h_j(i)]$
14:    **end for return**
    **end procedure**

---

## A.2 HIGHER-ORDER SKETCHES

While the count sketch is traditionally used to project vectors into lower-dimensional space, it is also possible to sketch higher order tensors. The advantage to sketching higher order tensors is two-fold for us: (1) The $\ell_2$ guarantees associated with vectorizing a tensor and then using a standard

count sketch does not scale well with increased modes and does not take advantage of the compact representations tensors offer. (2) In the forward pass discussion of Section 4, for multi-layer perceptrons, our parameterized weights imitate the count sketch of vectors (namely, the last layer's output), but for CNN layers, our parametrized weights imitate multiplying matrices by count sketch matrices (since the last layer's output is a matrix), which are not modeled under the usual count sketch model. A variety of tensor sketching algorithms exist Pham & Pagh (2013); Jin et al. (2019); Wang et al. (2015), but we elect to use the higher-order sketch (HCS) of Shi et al. (Shi & Anandkumar, 2019) due to its ease of implementation and resemblance to the count sketch.

We leave the details of the HCS algorithm for tensors with order 3 or greater to (Shi & Anandkumar, 2019), but for matrices, the method is straightforward. Let $W \in \mathbb{R}^{d \times d}$, for simplicity (but one may use rectangular matrices as well). First, in the same manner as the count sketch, draw $k$ pairwise-independent hash functions $h_i$ and sign maps $s_i$ of sketching length $c$. To sketch $W$, simply compute and maintain the collection of products $\hat{W}_i = H_i^\top H_i W$. The decompression (recovery) of $W$ is denoted $\hat{W}$, where coordinate-wise, we have that $(\hat{W})_{mn} = \text{median}\{\hat{W}_{i_{mn}}\}_{i=1}^k$, for $m, n \in [d]$.

### A.3 Two-Sided Sketching

Our discussion of the HCS in Section A.2 only allows compression along a single mode of a matrix $W \in \mathbb{R}^{d \times d}$, but it is possible to sketch along both modes of the matrix, allowing for further compression. In architectural terms, this allows us to decrease the width of each layer and decrease the density of inter-layer connectivity. The procedure is a simple extension of the one-sided HCS. Draw $k$ pairwise-independent hash functions $h_i$ and sign maps $s_i$. To sketch $W$, maintain the collection of products $\hat{W}_i = (H_{i_1}^\top H_{i_1}) W (H_{i_2}^\top H_{i_2})$, where we note that $H_{i_1}$ and $H_{i_2}$ are independently drawn. The decompression (recovery) of $W$ is denoted $\hat{W}$, where coordinate-wise, we have that $(\hat{W})_{mn} = \text{median}\{\hat{W}_{i_{mn}}\}_{i=1}^k$, for $m, n \in [d]$.

**Backpropagation Rule**  Following the single-sketch model of Section 4, we derive the backpropagation rule for a two-sided sketch. We first require a lemma, which is a basic result of matrix calculus:

**Lemma 2.** *Let* $X \in \mathbb{R}^{m \times n}$, $A \in \mathbb{R}^{p \times m}$, $B \in \mathbb{R}^{m \times q}$, *then*

$$\frac{\partial AXB}{\partial X} = B \otimes A^\top \tag{9}$$

*where* $\otimes$ *is the Kronecker product.*

We may now use the above result for calculating the gradient of our network after sketching our layers $\{W_t^\ell\}_{\ell=1}^L$ from both sides, in the same notation as the federated learning setup discussed in Section 2.

**Proposition 3.** *Let* $W_t^\ell \in \mathbb{R}^{d \times d}$, $H_1^\ell \in \mathbb{R}^{p \times d}$, $H_2^\ell \in \mathbb{R}^{d \times q}$. *Denote* $\tilde{W}_t^\ell = H_1^\ell W_\ell H_2^\ell$, *i.e., a weight matrix sketched from both sides. Then,*

$$\nabla_{W_t^\ell} \mathcal{L}(W_t^\ell, z) = (H_2^\ell \otimes H_1^{\ell^\top}) \nabla_{\tilde{W}_t^\ell} \mathcal{L} \tag{10}$$

*Proof.* By the chain rule,

$$\frac{\partial \mathcal{L}}{\partial W_t^\ell} = \frac{\partial \mathcal{L}}{\partial \tilde{W}_t^\ell} \frac{\partial \tilde{W}_t^\ell}{W_t^\ell} \tag{11}$$

$$= \frac{\partial \mathcal{L}}{\partial \tilde{W}_t^\ell} (H_1^\ell \otimes H_2^{\ell^\top}), \tag{12}$$

where the last line follows by Lemma 2. By transposing both sides of the equation to obtain gradient equations and noting that $(A \otimes B)^\top = B^\top \otimes A^\top$, the result follows. $\qquad \square$

## B  Proof of Theorem 1

Before proving our main convergence result, we define a recursive sequence crucial to error-feedback analysis following (Rothchild et al., 2020; Karimireddy et al., 2019).

Let $C(x) = \text{Top-}k(\mathcal{U}(H(x)))$. We define the temporal sequence $\tilde{W}_t = W_t - e_t - \frac{\eta\rho}{1-\rho}u_{t-1}$. This sequence is recursive:

$$
\begin{aligned}
\tilde{W}_t &= W_{t-1} - C(\eta u_{t-2} + g_{t-1}) \\
&\quad + C(\eta(\rho u_{t-2} + g_{t-1}) + e_{t-1}) \\
&\quad - \eta(\rho u_{t-2} + g_{t-1}) - e_{t-1} - \frac{\eta\rho}{1-\rho}z_{t-1} \\
&= W_{t-1} - e_{t-1} - \eta g_{t-1} - \eta\rho u_{t-2} \\
&\quad - \frac{\eta\rho}{1-\rho}(\rho u_{t-2} + g_{t-1}) \\
&= W_{t-1} - e_{t-1} - \frac{\eta\rho}{1-\rho}u_{t-2} - \frac{\eta}{1-\rho}g_{t-1} \\
&= \tilde{W}_{t-1} - \frac{\eta}{1-\rho}g_{t-1}
\end{aligned}
$$

This is an almost-stochastic SGD update, but we must prove $\nabla f(\tilde{W}_t) \approx \nabla f(W_t)$. To do so, we will provide two results which bound $\mathbb{E}||u_t||^2$ and $\mathbb{E}||e_t||^2$, respectively.

**Lemma 4.** $\mathbb{E}||u_{t-1}||^2 \leq \left(\frac{G}{1-\rho}\right)^2$.

*Proof.*

$$
\mathbb{E}||u_t||^2 = \mathbb{E}||\sum_{i=1}^{t}\rho^i g_i||^2 \leq \mathbb{E}||\sum_{i=1}^{\infty}\rho^i g_i||^2 \leq \left(\frac{G}{(1-\rho)}\right)^2. \tag{13}
$$

$\square$

**Proposition 5** (Karimireddy et al. (2019), Lemma 3). $\mathbb{E}||e_{t-1}||^2 \leq \frac{4(1-\epsilon^2)\eta^2 G^2}{\epsilon^2(1-\rho)^2}$.

*Proof.*

$$
\begin{aligned}
\mathbb{E}||e_{t+1}||^2 &= \mathbb{E}||\eta(\rho u_t + g_t) + e_t - C(\eta(\rho u_t + g_t) + e_t)||^2 \\
&\leq (1-\epsilon)\mathbb{E}||\eta(\rho u_t + g_t) + e_t||^2 \\
&\leq (1-\epsilon)\Big((1+\gamma)||e_t||^2 + (1+1/\gamma)\eta^2||u_t||^2\Big) \\
&\leq (1-\epsilon)\left(||e_{t-1}||^2 + \frac{(1+1/\gamma)\eta^2 G^2}{(1-\rho)^2}\right) \\
&\leq \sum_{i=0}^{\infty}\frac{((1-\epsilon)(1+\gamma))^i(1+1/\gamma)\eta^2 G^2}{(1-\rho)^2} \\
&\leq \frac{(1-\epsilon)(1+1/\gamma)\eta^2 G^2)}{1-((1-\epsilon)(1+\gamma))} \\
&\leq \frac{4(1-\epsilon)\eta^2 G^2}{\epsilon^2(1-\rho)^2},
\end{aligned}
$$

where in the third inequality, we use Young's inequality; in the fourth inequality, we invoke Lemma 4; and in the last line, we bound everything by choosing $\gamma = \frac{\epsilon}{2(1-\gamma)}$. $\square$

We will now prove our convergence result.

*Proof of Theorem 1.* Consider a Comfetch model weight: $w$ is sent down by the server as a sketch key pair $\{\mathcal{U}_H(\cdot), H(w)\}$ where $H$ is a count sketch structure of size $\mathcal{O}\left(\frac{1}{\epsilon}\log\frac{d}{\delta}\right)$ (following notation as described in Section 2.3). The objective function of our sketch network is $\tilde{f}(w) \triangleq f(\mathcal{U}_H(H(w)))$.

Although $\mathcal{U}_H(Hw)$ is not a linear transformation, we note that for any fixed single count sketch matrix $H^*$, $||H^\top Hw|| \leq \frac{d}{c}||w||$, therefore, under the count sketch median recovery scheme, $||\mathcal{U}_H(H(w))|| \leq \frac{d}{c}||w||$. Now, invoking Assumption 1, we have that

$$
\begin{aligned}
||\nabla \hat{f}(x) - \nabla \tilde{f}(y)|| &= ||\nabla f(\mathcal{U}_H(H(x))) - \nabla f(\mathcal{U}_H(H(w)))|| \\
&\leq L||\mathcal{U}_H(H(x)) - \mathcal{U}_H(H(y))|| \\
&\leq \frac{Ld}{c}||x - y||.
\end{aligned}
$$

This establishes that $\tilde{f}$ is $\frac{Ld}{c}$-smooth. Furthermore, Assumption 2 trivially holds for the stochastic gradients of $\hat{f}$ as well. Denote $\tilde{w}_t = \mathcal{U}_H(H(w))$. We follow analysis in Rothchild et al. (2020); by $\frac{Ld}{c}$-smoothness of $\tilde{f}$,

$$
\begin{aligned}
\mathbb{E}\tilde{f}(w_{t+1}) &\leq \tilde{f}(\tilde{w}_t) + \left\langle \nabla \tilde{f}(\tilde{w}_t), \mathbb{E}[\tilde{w}_{t+1} - \tilde{w}_t] \right\rangle + \frac{Ld}{2c}\mathbb{E}\left|\left| \tilde{w}_{t+1} - \tilde{w}_t \right|\right|^2 \\
&\leq \tilde{f}(\tilde{w}_t) + \left\langle \nabla \tilde{f}(\tilde{w}_t), \mathbb{E}[\tilde{w}_{t+1} - \tilde{w}_t] \right\rangle + \frac{Ld\eta^2}{2c(1-\rho)^2}\mathbb{E}||\tilde{g}_t||^2 \\
&\leq \tilde{f}(\tilde{w}_t) - \frac{\eta}{(1-\rho)}\left\langle \nabla \tilde{f}(\tilde{w}_t), \mathbb{E}[g_t] \right\rangle + \frac{Ld\eta^2}{2c(1-\rho)^2}\mathbb{E}||\tilde{g}_t||^2 \\
&\leq \tilde{f}(\tilde{w}_t) - \frac{\eta}{1-\rho}\langle \nabla \tilde{f}(\tilde{w}_t), \nabla \tilde{f}(w_t)\rangle + \frac{Ld\eta^2 G^2}{2c(1-\rho)^2} \\
&= \tilde{f}(\tilde{w}_t) - \frac{\eta}{(1-\rho)^2}\mathbb{E}||\nabla \tilde{f}(w_t)||^2 + \frac{\eta}{2(1-\rho)^2}\mathbb{E}||\nabla \tilde{f}(w_t)||^2 \\
&\quad + \frac{\eta}{2(1-\rho)^2}\mathbb{E}||\nabla \tilde{f}(\tilde{w}_t) - \nabla \tilde{f}(w_t)||^2 + \frac{Ld\eta^2 G^2}{2c(1-\rho)^2} \\
&\leq f(\tilde{w}_t) - \frac{\eta}{2(1-\rho)}\mathbb{E}||\nabla \tilde{f}(w_t)||^2 + \frac{\eta L^2 d^2}{2(1-\rho)c^2}||\tilde{w}_t - w_t||^2 + \frac{Ld\eta^2 G^2}{2c(1-\rho)^2} \\
&\leq f(\tilde{w}_t) - \frac{\eta}{2(1-\rho)}\mathbb{E}||\nabla \tilde{f}(w_t)||^2 + \frac{\eta L^2 d^2}{2(1-\rho)c^2}||e_t + \frac{\eta\rho}{1-\rho}u_{t-1}||^2 + \frac{Ld\eta^2 G^2}{2c(1-\rho)^2}
\end{aligned}
$$

.

We must bound $||e_t + \frac{\eta\rho}{1-\rho}u_{t-1}||^2$ now. However, since we only maintain $H(e_t)$ and $H(u_{t-1})$, we may instead consider the sketched norm,

$$
||H(e_t) + \frac{\eta\rho}{1-\rho}H(u_{t-1})||^2.
$$

By size of $H(\cdot)$ and Assumption 3, we have with probability $1 - \delta$ that our sketch will recover all $(\ell_2, \epsilon)$-heavy hitters of $w$ and that $||H(w)|| \leq (1 + \epsilon)||w||$ (Cormode & Muthukrishnan, 2005). Invoking Lemma 4,

$$
||H(u_{t-1})||^2 \leq \Big(\sum_{i=1}^{t-1} \rho^i ||H(\tilde{g}_i)||^2\Big) \leq \Big(\sum_{i=1}^{t-1} \rho^i (1+\epsilon)G\Big)^2 \leq \Big(\frac{(1+\epsilon)G}{1-\rho}\Big)^2. \tag{14}
$$

We also have by Proposition 5 that

$$
||H(e_t)||^2 \leq \frac{(1+\epsilon)^2(1-\epsilon)(1+1/\gamma)\eta^2 G^2}{1 - ((1-\epsilon)(1+\gamma))}. \tag{15}
$$

By choosing $\gamma = \frac{\epsilon}{2(1-\epsilon)}$ in Equation 15, we have that

$$
||H(e_t)||^2 \leq \frac{4(1+\epsilon)^2(1-\epsilon)\eta^2 G^2}{\epsilon^2(1-\rho)^2}. \tag{16}
$$

Using Equations 16 and 14 to upper bound $||e_t + \frac{\eta\rho}{1-\rho}u_{t-1}||^2$, we conclude that

$$\mathbb{E}||\nabla\tilde{f}(w_t)||^2 \leq \frac{2(1-\rho)}{\eta}\big(\tilde{f}(\tilde{w}_t) - \mathbb{E}\tilde{f}(\tilde{w}_{t+1})\big)$$

$$+ \frac{2(1-\rho)}{\eta}\Big(\frac{4\eta L^2 d^2(1+\epsilon)^2\eta^2 G^2}{2(1-\rho)c^2(1-\epsilon)\epsilon^2(1-\rho)^2}\Big)$$

$$+ \frac{2(1-\rho)}{\eta}\Big(\frac{Ld\eta^2 G^2}{2c(1-\rho)^2}\Big).$$

Averaging over $T$ and setting $\eta = \frac{c(1-\rho)}{2Ld\sqrt{T}}$ gives us our result. $\qquad\square$

*Remark.* The theory is more restrictive than our empirical results, which is often the case with sketching compression schemes, as the classical sketching concentration bounds are ensured via multiple sketches. In particular, using only a single sketch works very well as shown in Section 6.

## C  MULTI-SKETCH COMFETCH

In this section, we describe how to incorporate the usage of multiple sketches per layer for `Comfetch`, which is described by Algorithm 3.

---

**Algorithm 3** Multi-sketch `Comfetch`

---

**Require:** initial weights $\{W_0^\ell\}_{\ell=1}^L$, learning rate $\eta$, number of iterations $T$, momentum parameter $\rho$, batch size $M$ of data, batch size $N$ of clients
    Init momentum term $\{u_0^\ell = 0\}_{\ell=1}^L$
2:  Init error accumulation term $e_0 = 0$
    **for** $t = 1, 2, \ldots, T$ **do**
4:       Init sketching and unsketching procedures $\{\mathcal{U}^\ell, \mathcal{S}^\ell\}_{\ell=1}^L$
       Uniformly select at random $N$ clients $c_1, c_2, \ldots, c_N$
6:       **loop** {in parallel on clients $\{c_i\}_{i=1}^N$}
        Download parameterized weight pairs $\{(\mathcal{U}^\ell, \mathcal{S}^\ell(W_t^\ell)\}_{\ell=1}^L$
8:       **for** $\ell = 1, 2, \ldots, L$ **do**
        Compute grads $g_i^\ell = \{\nabla_{H_j W_t}\mathcal{L}(R_i^\ell W_t^\ell, z \sim \mathcal{D}_i)\}_{j=1}^k$      $\triangleright$ See equation 24
10:     **end for**
       Send $\{g_t^\ell\}_{\ell=1}^L$ to Central Server
12:    **end loop**
     **for** $\ell = 1, 2, \ldots, L$ **do**
14:      Aggregate restored gradients: $g_t^\ell = \frac{1}{N}\sum_{i=1}^N \mathcal{G}(g_i^\ell)$      $\triangleright$ See equation 25
       Update sketched momentum: $u_t^\ell = \rho u_{t-1}^\ell + g_t^\ell$
16:      Update error feedback: $e_t = \eta u_t + e_t$
       Approximate gradient with feedback: $\Delta^t = \text{Top-k}(e_t)$
18:      Error accumulation: $e^{t+1} = e_t - \Delta_t$
       Update weights: $W_{t+1} = W_t - \Delta_t$
20:    **end for**
   **end forreturn** $\{w_T^\ell\}_{\ell=1}^L$

---

### C.1  MODEL TRANSMISSION AND DOWNLOAD

At iteration $t$, the central server first prepares the global model for the transmission by sketching down all the current weights $\{W_t^\ell\}_{\ell=1}^L$, . We assume that our layers are either convolutional or fully-connected as described in Section 2, and for simplicity, that all our $W_t^\ell$ are of size $d \times d$, but they can be rectangular in practice. For each weight $W_t^\ell$, the central server randomly draws count sketch matrices $\{H_i^\ell\}_{i=1}^k$, where $H_i^\ell \in \mathbb{R}^{c \times d}$, $c << d$ is the sketching length, and transmits $\{(H_i^{\ell\top}, H_i^\ell W_t^\ell)\}_{i=1}^k$ to a selection of $N$ uniformly randomly drawn

clients, who then download these sketched parameters into local memory. Henceforth, will denote $\{(\mathcal{U}^\ell, S^\ell(W_t^\ell)\} \triangleq \{(H_i^{\ell\top}, H_i^\ell W_t^\ell)\}_{i=1}^k$ for $\ell \in [L]$.

**Cost Complexity** We note that $k$ corresponds to the number of independent sketches, which is required theoretically to achieve a guarantee on approximating $W_t^\ell$, but in practice, we observe in Section 6 that one sketch is sufficient for model convergence. Also note that any $H_i^\ell$ bijectively corresponds to a hash function $h_i^\ell : d \rightarrow c$ which is representable as a length $d$ vector. Hence, in practice, the central server will transmit $\{(h_i^\ell, H_i^\ell W_t^\ell)\}_{i=1}^k$, for a total local memory and transmission cost of $\mathcal{O}\big((kc + 1)d\big)$, which in the one-sketch ($k = 1$) case is less than the $O(d^2)$ cost of transmitting the full weight.

## C.2 CLIENT UPDATE

The client $C_i$ will now conduct a single round of training on the sketched network parameters using their local data. Often in practice, the client distributions $\mathcal{D}_i$ will be finite and small Kairouz et al. (2019), so we assume that the client is always taking the full gradient with respect to the weights, but the algorithm generalizes to stochastic gradients as well.

**Forward pass** Let $x \in \mathcal{D}_i$ and let $\{(\mathcal{U}^\ell, S^\ell(W_t^\ell)\} \triangleq \{(H_i^{\ell\top}, H_i^\ell W_t^\ell)\}_{i=1}^k$ for $l \in [L]$, where $L$ is the depth of our network. Following the notations described in Section 2, the forward pass of a fully-connected layer is

$$x^\ell = \sigma(\mathcal{U}^\ell(S^i(W_t^\ell x^{i-1})), 1 \leq \ell \leq L - 1 \tag{17}$$

$$\hat{y} = \mathbf{a}^\top x^L, \tag{18}$$

where $(\mathcal{U}^\ell(S^\ell(W_t^\ell))x)_i \triangleq \underset{1 \leq j \leq k}{\text{median}}\{(H_j^{\ell\top} H_j^\ell W_t^\ell x\}$ for any $x \in \mathbb{R}^d$, in agreement with the count sketch procedure of Algorithm 2. Similarly, for a convolutional ResNet, we have that

$$x^1 = \sqrt{\frac{c_\sigma}{m}}\sigma\Big(\mathcal{U}^1(\mathcal{S}^1(W_t^1))\phi(x^0)\Big) \tag{19}$$

$$x^\ell = x^{\ell-1} + \frac{c_{res}}{L\sqrt{m}}\sigma\Big(\mathcal{U}^\ell(\mathcal{S}^\ell(W_t^\ell))\phi_\ell(x^{\ell-1})\Big),$$

$$2 \leq \ell \leq L \tag{20}$$

$$\hat{y} = \langle W_t^L, x^L \rangle, \text{ where } W^L \in \mathbb{R}^{m \times p}. \tag{21}$$

*Remark 1:* We never directly compute a $d \times d$ weight matrix at any stage.
*Remark 2:* In the convolutional case, where the inputs between layers are matrices, $H_i W x = H_i(W x)$ can be regarded as a higher-order count sketch (HCS) of $W x$ as described in Appendix A.

**Backward pass** In order to compute the gradient, we must first clearly define the weights of the modified network. Let $\{H_i\}_{i=1}^k$ be a random set of $c \times d$ count sketch matrices and let $x \in \mathbb{R}^d$. We have that $\hat{x}_i := \mathcal{U}(\mathcal{S}(x))_i = \underset{1 \leq j \leq k}{\text{median}}\{(H_j^\top H_j x)_i\}$. If sketch $H_{j_i}$ results in the median recovery of the $i$th coordinate of $x$, then we have the following representation:

$$\hat{x} = \sum_{i=1}^d E_i H_{j_i}^\top H_{j_i} x, \tag{22}$$

where $E_i \in \mathbb{R}^{d \times d}$ is a matrix with 1 at entry $i, i$ and 0 everywhere else. Therefore, it possible to define $\hat{x} = \mathcal{U}(\mathcal{S}(x)) = Ax$ where $A \in \mathbb{R}^{d \times d}$. That is, we can represent the count sketch recovery of $x$ as a matrix transformation.

By the above discussion, we can represent $\mathcal{U}^\ell(\mathcal{S}^\ell(W_t^\ell x^{\ell-1})) = R_i^\ell W_t^\ell x^{\ell-1}$, where $R_i^\ell \in \mathbb{R}^{d \times d}$ is referred to as the *recovery matrix*. Hence, the weights of the client network are $R_i^\ell W_t^\ell$. Note that we specifically indicate the client index $i$, since the recovery matrix will vary depending on the local data.

Now that we have have defined the weights of our client models, we may now take gradients. The server will want to receive $\frac{\partial \mathcal{L}(R_i^\ell W_t^\ell, z)}{\partial W_t^\ell}$ as an approximation of $\frac{\partial \mathcal{L}(W_t^\ell, z)}{\partial W_t^\ell}$, but the client will not want to compute $\frac{\partial \mathcal{L}(R_i^\ell \hat{W}_t^\ell, z)}{\partial W_t^\ell}$, since it is of size $d \times d$. We will want to transmit a $\mathcal{O}(kc \times d)$ packet of data which will allow the

server to compute $\frac{\partial \mathcal{L}(R_i^\ell W_t^\ell, z)}{\partial W_t^\ell}$. To this end, let $\{H_i^\ell\}_{i=1}^k$ be the set of count sketch matrices associated with layer $\ell$. Let $E_{H_i} \in \mathbb{R}^{d \times d}$ denote the matrix which has a 1 at entry $j, j$ for $1 \leq j \leq d$ if $H_i$ contains the median recovery of $(W_t^\ell x^{\ell-1})_j$ and 0 everywhere else. We have then that,

$$R_i^\ell = \sum_{i=1}^k E_{H_i} H_i W_t^\ell x^{\ell-1}. \tag{23}$$

Therefore, by the chain rule of the total derivative,

$$\frac{\partial \mathcal{L}(R_i^\ell W_t^\ell, z)}{\partial W_t^\ell} = \sum_{i=1}^k \frac{\partial \mathcal{L}(R_i^\ell W_t^\ell, z)}{\partial H_i^\ell W_t^\ell} \frac{\partial H_i^\ell}{\partial W_t^\ell}. \tag{24}$$

Thus, the client will upload $g_t^\ell \triangleq \{\nabla_{H_i W_t^\ell} \mathcal{L}(R_i^\ell W_t^\ell, z)\}_{i=1}^k$ for all $\ell \in [L]$.

*Remark.* We subtly avoided the point that the recovery matrix $R_i^\ell$ as described is recursively determined by the initial input. Therefore, our sketched weights will not be represented as simple linear transformations of the original weights. However, since the client will clearly be determining gradients through an `autograd`-like library, this will not pose an issue anyways.

**Cost Complexity**   If the client chooses to upload the gradients contained within $g_t^\ell$ in a predefined manner (for example, in the order the sketches were transmitted), then the central server will know which gradients correspond to which sketch, and thus will be able to compute equation without any additional information. Thus, the total communication cost is $\mathcal{O}(kcd)$, which in the single-sketch $k = 1$ scenario, is a strong improvement over the usual uplink cost of $\mathcal{O}(d^2)$ and even cheaper than the download cost.

*Remark.* The complexity of computing $R_i^\ell W_t^\ell x^{\ell-1}$ is $\mathcal{O}(2kcd + k)$, since the client must individually compute $H_i^\top H_i W_t x_t^{\ell-1}$ for all $i \in [k]$ to determine the median coordinates. This may not be cheaper than the usual $\mathcal{O}(d^2)$ matrix-vector multiplication of the original layer if too many sketches are used, but as we demonstrate in Section 6, a single sketch is sufficient.

### C.3   MODEL UPDATE

The Central Server aggregates the $\{g_i^\ell\}_{\ell=1}^L$ across all $i \in [N]$. We describe the procedure for updating the weight of a fixed layer $\ell$. For each $i \in [N]$, the server computes

$$\mathcal{G}(g_i^\ell) = \sum_{j=1}^k H_j^{\ell \top} \nabla_{H_j^\ell W_t^\ell} \mathcal{L}(R_i^\ell W_t^\ell, z). \tag{25}$$

The server takes an average over the $\mathcal{G}(g_\ell)$ to compute a stochastic gradient of $f(R^\ell W_t^\ell)$ with respect to $W_t^\ell$. That is,

$$\nabla_{W_t^\ell} f(R_i^\ell W_t^\ell) = \mathbb{E} \frac{1}{N} \sum_{i=1}^N \mathcal{G}(g_i^\ell). \tag{26}$$

The remainder of the model update follows the error-feedback and momentum scheme of `FetchSGD` Rothchild et al. (2020). Error-feedback allows for the correction of error associated with gradient approximations. In the case of `FetchSGD`, error-feedback corrects the error associated with a taking a sketch and unsketch of the gradients. In the case of `Comfetch`, we are correcting the error associated with using $\nabla_{W_t^\ell} f(R^\ell W_t^\ell)$ as an approximation of $\nabla f(W_t^\ell)$. The reader is encourage to consult the work of Karimireddy et al. and Stitch et al. for further details on error-feedback for SGD-like methods Karimireddy et al. (2019); Stich et al. (2018).

*Remark.* We subtly avoided the point that the recovery matrix $R_i^\ell$ as described is recursively determined by the initial input. Therefore, our sketched weights will not be represented as simple linear transformations of the original weights. However, since in practice the client will clearly be determining gradients through an `autograd`-like library, this will not pose an issue.

## D   ADDITIONAL LANGUAGE TASK DATA

In this section, we present additional data in Table 3 and Figure 5 for language tasks not included in the main paper. The results demonstrate that our `Comfetch` preserves accuracy while reducing the size of weights in gates of the LSTM layer by different ratios. We maintain a high test accuracy while reducing the number of parameters from 9607 to 2439. We exclude embedding parameters from the overall parameter counts as they serve as inputs and can be pre-trained.

| Method | Bandwidth Compression | Memory Compression | Number of Params | Test Acc (%) |
|---|---|---|---|---|
| FedAvg | 1 | 1 | 9607 | 47.58 |
| FedAvg-1/2 | 1 | 1/2 | 4423 | 49.78 |
| FedAvg-1/4 | 1 | 1/4 | 2599 | 44.01 |
| FedAvg-1/8 | 1 | 1/8 | 1879 | 25.63 |
| FetchSGD | 1 | 1 | 9607 | 86.81 |
| Comfetch | 1 | 1 | 9607 | 80.44 |
| Comfetch-1/2 | 1/2 | 1/2 | 5511 | 81.50 |
| Comfetch-1/4 | 1/4 | 1/4 | 3463 | 80.57 |
| Comfetch-1/8 | 1/8 | 1/8 | 2439 | 78.61 |

**Table 3:** Model accuracies under different memory footprints in clients, for predicting part of speech taggings for MNLI Williams et al. (2017) sentences using LSTM. We exclude embeddings from parameter counts as they serve as inputs and can be pretrained.

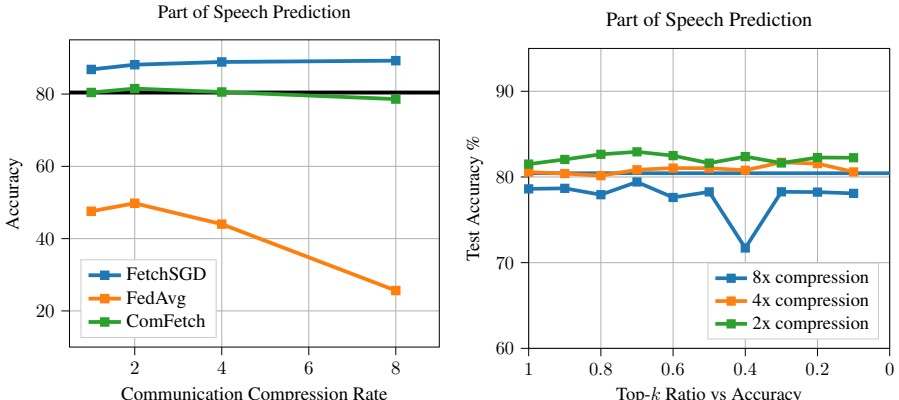

**Figure 5:** Test accuracy achieved on predicting part of speech taggings for MNLI Williams et al. (2017) sentences. On the left, we present quantitative comparisons to other methods. The horizontal line reflects the simple baseline where no compression is applied during training. On the right, we report the test accuracy with different compression rate while varying the $K$ ratio. FetchSGD only compresses the network weights during communication, but each client still needs to decompress the entire network locally to perform training.

# E   PREDICTION ERROR BOUND

In this section, we provide a bound on the error between the prediction of a fully-connected multi-layer network and its sketched counterpart. We denote by $\phi : \mathbb{R}^d \to \mathbb{R}^d$, the ReLU (rectified linear unit), where for any $x \in \mathbb{R}^d$, we have that $(\phi(x))_i = \max((x)_i, 0)$. For ease (and an abuse) of notation, we let $\psi \circ W_L \circ \psi \circ W_{L-1} \circ \cdots \circ \psi \circ W_1 x = \phi(W_L(\phi(W_{L-1}(\cdots (\phi(W_1 x) \cdots )))$, where $W_\ell \in \mathbb{R}^{d \times d}$ for $1 \le \ell \le L$, $x \in \mathbb{R}^d$, and we are freely allowing $W_1$ to act as both a linear transformation and a matrix multiplication (i.e., $W_1 \circ x = W_1 x$).

**Theorem 6.** *Let*

$$\hat{y}_L = \psi \circ W_L \circ \psi \circ W_{L-1} \circ \cdots \circ \psi \circ W_1 \circ x, \tag{27}$$

$W_\ell \in \mathbb{R}^{d \times d}$ for $1 \le \ell \le L$, $x \in \mathbb{R}^d$,, and $\psi$ is the ReLU activation function. Now let

$$\tilde{y}_L = \psi \circ H_L^{-1} H_L W_L \circ \psi \circ H_{L-1}^{-1} H_{L-1} W_{L-1} \circ \tag{28}$$

$$\cdots \circ \psi \circ H_1^{-1} H_1 W_1 x, \tag{29}$$

where $H_\ell^{-1} H_\ell W_i$ reflects a count sketch recovery of $W_\ell$. If for each $H_\ell^{-1} H_\ell W_\ell$ we have chosen each sketching length of $W_\ell$ as $c = \Omega(\|W_\ell\|_F^2 / \epsilon^2)$ and the number of independent sketches as $\mathcal{O}(\log \frac{d}{\delta})$, for $0 < \delta < 1$, then with $(1 - \delta)^L$ probability we have that

$$\|\tilde{y}_L - \hat{y}_L\| \le \sum_{j=1}^{L} g_j(x), \tag{30}$$

where $g_j(x) = \lambda_j \lambda_{j+1} \cdots \lambda_L \|x\| d^2 \epsilon^2 \prod_{n=1}^{j-1} \hat{\lambda}_n$, $\lambda_i$ is the maximum singular value of $W_i$, and $\hat{\lambda}_i$ is the maximum singular value of $H_\ell^{-1} H_\ell$. We let $g_0 = d^2 \epsilon^2 \|x\|$.

*Proof.* We proceed by induction on the number of layers $L$. For simplicity, we denote $\hat{W}_i := H_i^{-1} \circ H_i \circ W_i$. For $L = 1$, we have that

$$||\psi \circ \hat{W}_i x - \psi W_i x|| \leq ||\hat{W}_i - W_i|| ||x|| \leq d^2 \epsilon^2 ||x|| = g_0, \tag{31}$$

where the first inequality follows by the fact the ReLU $\psi$ is 1-Lipschitz and the second inequality follows by the conventional HCS guarantee (Shi & Anandkumar, 2019) and our prescribed width and depth of sketches, $\mathcal{O}(\frac{1}{\epsilon^2} \log \frac{d}{\delta})$. Assume the hypothesis holds for $L = k$ layers, then for $L = k + 1$ layers we have that

$$||\tilde{y}_{k+1} - \hat{y}_{k+1}|| = ||\psi \circ \hat{W_{k+1}} \circ \tilde{y}_k - \psi \circ W_{k+1} \circ \hat{y}_k|| \tag{32}$$

$$\leq ||\hat{W}_{k+1} \circ \tilde{y}_k - W_{k+1} \circ \hat{y}_k|| \tag{33}$$

$$\leq ||\hat{W}_{k+1} \circ \tilde{y}_k - W_{k+1} \circ \tilde{y}_k + W_{k+1} \circ \tilde{y}_k - W_{k+1} \circ \hat{y}_k|| \tag{34}$$

$$\leq ||\hat{W}_{k+1} \circ \tilde{y}_k - W_{k+1} \circ \tilde{y}_k|| + ||W_{k+1} \circ \tilde{y}_k - W_{k+1} \circ \hat{y}_k|| \tag{35}$$

$$\leq ||\hat{W}_{k+1} - W_{k+1}|| ||\tilde{y}_k|| + ||W_{k+1}|| ||\tilde{y}_k - \hat{y}_k|| \tag{36}$$

$$\leq d\epsilon^2 ||x|| \prod_{n=1}^{k} \hat{\lambda}_n + \lambda_{k+1} \sum_{j=1}^{k} g_j \tag{37}$$

$$= \sum_{j=1}^{k+1} g_j, \tag{38}$$

where the second to last inequality follows by applying the inductive hypothesis to the right term $||\tilde{y}_k - \hat{y}_k||$ and noting that for the left term,

$$||\tilde{y}_k|| = ||\psi \circ \hat{W}_k \circ \psi \hat{W}_{k-1} \circ \cdots \circ \hat{W}_i x|| \tag{39}$$

$$\leq ||x|| \prod_{i=1}^{k} ||\hat{W}_i|| \leq \hat{\lambda}_i ||x||. \tag{40}$$

The probabilistic guarantee of $(1 - \delta)^L$ follows by the independence of each individual layer sketching. □

The above result theoretically demonstrates that the noise is controllable via increased sketched length and number of independent sketches, and in general, requires increased space complexity as $L$ increases.

### E.1    MULTI-SKETCH ABLATION & RESNET-9 EXPERIMENTS

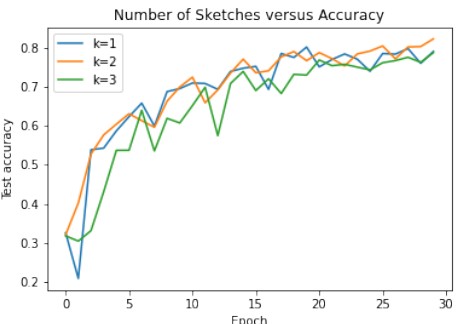

**Figure 6: Multi-Sketch Comfetch.** We assess the affect of using multiple sketches during training, for a single client training of CIFAR-10 with $87.5\%$ compression. The use of multiple sketches has no noticeable effect on model performance.

Theorem 1 and nearly all of count sketch theory use multiple sketches to obtain convergence guarantees. To assess the effects of using using multiple sketches on model performance, in Figure 6, we train a single client on CIFAR-10 using varying number of sketches between layers for weight recovery. We set our compression rate to $87.5\%$ since Comfetch models at this level of compression experience noticeable performance decline. We find that using multiple sketches does not improve performance, justifying usage of a single sketch in our experiments. We believe that one-sketch guarantees would be valuable for uplink/downlink compression literature since the single-sketch compression strategy is successful in practice.

### E.2   1 CLIENT, RESNET-9

In Table 4, we conduct a simple study examining how Comfetch compression affects single (unfederated) client training when ResNet-9 is used to train over CIFAR-10 over 25 epochs. We additionally conduct random pruning of a non-sketched model to mimic compression. Fixed random pruning to achieve the same compression amount outputs worse models. It is important to note that pruning works well for pre-trained models, but this study demonstrates the ineffectiveness of one-time pruning prior to training. Figure 2 demonstrates the training and test curves.

| Method | Model Size | Test Acc (%) |
|---|---|---|
| No sketch | 1 | 86.04 |
| No sketch | 1/2 | 75.38 |
| Comfetch | 1/2 | **87.89** |
| No sketch | 1/4 | 73.34 |
| Comfetch | 1/4 | **86.12** |
| No sketch | 1/8 | 72.40 |
| Comfetch | 1/8 | **81.18** |

**Table 4:** Test accuracy under different memory footprints in clients for the CIFAR-10 Krizhevsky et al. (2009) image classification task. Comfetch maintains a large and powerful global model under communication compression and client memory compression. The no sketch model is compressed via random pruning, which performs much worse than our sketch-compressed models.

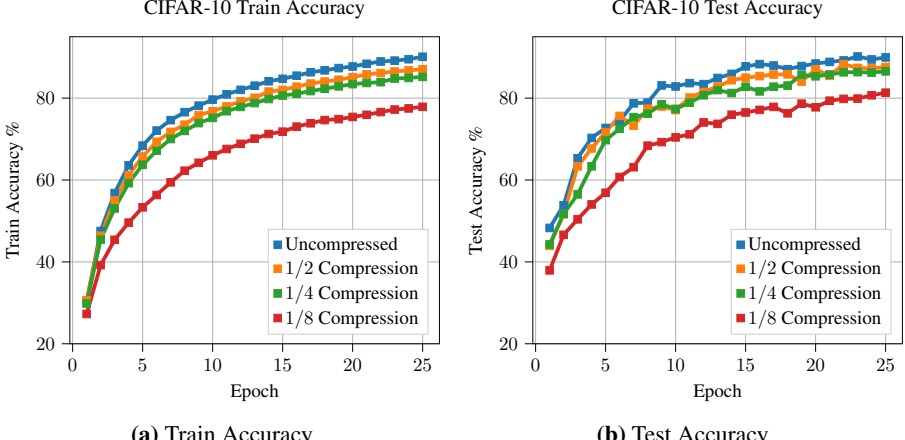

(a) Train Accuracy          (b) Test Accuracy

**Figure 7:** Test accuracy convergence of 1-client ResNet-9 Comfetch under varying compression rates. (a)-(b) correspond to the CIFAR-10 Krizhevsky et al. (2009) image classification. Similar accuracy with 1) different Comfetch compression rates suggests that our method retains the expressive power of the model while reducing the parameter sizes.

