# OpenReview forum: "Comfetch: Federated Learning of Large Networks on Constrained Clients via Sketching"
_ICLR.cc/2024/Conference — ICLR 2024 Conference Withdrawn Submission_

### Official Review · Reviewer_qrJM · 2023-10-24

**Soundness:** 2 fair
**Presentation:** 3 good
**Contribution:** 2 fair
**Rating:** 3
**Confidence:** 4

**Summary:**

This work proposes a new FL algorithm, Comfetch, which reduces communication, computation, and memory costs when training large models.
By using the count sketch algorithm, Comfetch attains a reduced representation of the global model and sends it to a client.
Owing to the reduced model size, server-client communication is reduced. Furthermore, the reduced model is also easier to be trained on resource-constrained clients, thereby alliviate memory and computation pressure on the client side.
The authors provide a convergence guarantee for the proposed algorithm and also show its effectiveness in CIFAR10/100 vision tasks.

**Strengths:**

This paper targets an important problem in federated learning: large model training for resource-constrained clients. Several contributions are highlighted as below

**1**, The proposed algorithm not only aims at reducing communication but also addressing *downlink complexity* (memory and computation costs in local training).

As the paper stated, there are many prior works targeting communication only. However, for an end-to-end solution, it is also crucial to address the complexities of local training.

**2**, The authors give a convergence analysis for the proposed algorithm.

I am glad that the authors show Comfetch converges under certain conditions. Without the convergence analysis, it is difficult to believe Comfetch can converge as there is no guarantee that the reduced representation via the count sketch approximates the original model.
Based on Assumption 3 and Theorem 1, the readers can at least see how the heavy hitter parameters affect convergence.

**Weaknesses:**

While I believe this work targets an important problem in FL, I still have several main concerns to be resolved.

**1**, The authors need to discuss more related work in the intro before motivation Comfetch.

In the intro, the authors only discussed communication-efficient algorithms, and ignored other methods that share a similar objective as Comfetch.
For instance, HeteroFL [1] and FjORD [2] also address *downlink complexity* by training sub-models on clients. I noticed the authors discuss these works in Section 3.2. However, it is more important to discuss these works in the intro and motivate why Comfetch is needed.
More importantly, even in terms of retraining dimensionality, PriSM [3] already designed a sub-model training method that can finally retrain a full model on the server side.
Therefore, the authors need to properly discuss these works in the intro.

**2**, The claim, that *preserving the full-dimensionality to maximize performance*, is not correct.

The authors imply that *full-dimensionality* is the key to preserving model performance (end of Sec 3.2). However, I do not think the claim holds for Comfetch.
In fact, in each round, Comfetch compresses the model with the reduced representation, thereby causing some information during training to be discarded. While the reconstructed model has the same dimension as the full model, it does not mean Comfetch can preserve the full model's performance. The reason is that, the resulting model can be very low-rank given small $c$, thereby significantly reducing the model's performance.
As a result, I do not think the motivation of Comfetch is well-supported.

**3**, There is no direct analysis of how good the client model is compared to the full model.

In Figure 1, the authors show a sub-model of a client. I believe $H^T(H\cdot W\cdot x)$ approximate $W \cdot x$ in the full model. The question is, how good this approximation is?
If the resulting sub-model is very different from the full model, will clients learn totally different models that may be useless to the server?
Low-rank based methods such as PriSM [3] at least guarantee each sub-model is a low-rank approximation to the full model. In Comfetch, I did not see how the approximation is bounded.

**4**, The evaluations are limited.

There are several limitations in the evaluations.

4.1, The authors only tested ResNet-18 on CIFAR10/100. As Comfetch targets a practical solution for training large models, it should be tested on more models and datasets. It is hard to convince readers that Comfetch is effective with only ResNet-18 on CIFAR.

4.2, Only 4 or 10 clients are simulated, which is an impractical setup. The authors should simulate more clients and investigate how Comfetch's performance scales with the number of clients.

4.3, Missing baselines. Priors HeteroFL [1], FjORD [2], and PriSM [3] all have the same objective as Comfetch. The authors should compare these methods, and evaluate how effective the count sketch method is. Importantly, as the authors claim *preserving the full-dimensionality to maximize performance*, it is important to compare Comfetch with HeteroFL and FjORD, and see if these methods with reduced dimensionality affect the final model's performance.

----

[1] Diao. "HeteroFL: Computation and Communication Efficient Federated Learning for Heterogeneous Clients". ICLR'20.

[2] Horvath, S. "Fjord: Fair and accurate federated learning under heterogeneous targets with ordered dropout". NeuIPS'21.

[3] Niu, Yue. "Federated Learning of Large Models at the Edge via Principal Sub-Model Training." TMLR'23.

**Questions:**

None

---

### Official Review · Reviewer_W1rB · 2023-11-03

**Soundness:** 2 fair
**Presentation:** 3 good
**Contribution:** 2 fair
**Rating:** 5
**Confidence:** 4

**Summary:**

The paper proposes a new federated learning (FL) framework that compresses the model updates via count-sketch algorithm to reduce the communication cost both in client-to-server and server-to-client communication. By de-sketching the low-dimensional gradients layer by layer as the forward pass progresses, the proposed method also reduces the computational cost. The paper presents empirical comparisons against "No Compression" case and L1 and random gradient sparsification methods.

**Strengths:**

Reducing the computational and communication costs simultaneously is an important problem in FL.

The paper provides a converge guarantee for the proposed method.

**Weaknesses:**

- The proposed method applies the count-sketch algorithm to FL which was previously done by many papers such as [1, 2]. The paper's main contribution seems to be in the idea that the clients do not need to de-sketch the whole compressed vector at once and instead, they can do it layer by layer as they go in the forward-pass. This way, the computational cost of de-sketching the whole gradient vector is avoided. This is a helpful mitigation but I feel that its technical novelty is limited.

- The empirical comparisons are very limited. In Table 1 and Figures 2-3, the proposed method is only compared against the "No compression" baseline which naturally yields better results than the proposed method. So, I am not sure how informative this comparison is about the efficacy of the proposed approach. In Figure 4, there is comparison against L1-based and random gradient sparsification methods. While it is good to have these comparisons in the paper, I wonder why the authors did not provide any comparison against other sketching-based FL compression papers [1, 2].

[1] Rothchild, Daniel, et al. "Fetchsgd: Communication-efficient federated learning with sketching." International Conference on Machine Learning. PMLR, 2020.

[2] Ivkin, Nikita, et al. "Communication-efficient distributed SGD with sketching." Advances in Neural Information Processing Systems 32 (2019).

**Questions:**

see weaknesses.

---

### Official Review · Reviewer_cTtw · 2023-11-04

**Soundness:** 4 excellent
**Presentation:** 4 excellent
**Contribution:** 3 good
**Rating:** 6
**Confidence:** 4

**Summary:**

This paper developed a sketching-based method for compressing both upload and download cost in federated learning. The authors provide a detailed description of their implementation, which is the differentiator and enablers for reduced download cost. They developed some theoretical analysis for the method under the error-feedback framework. The experiments show that the proposed method is effective even under high compression ration, and outperforms L1-magnitude pruning and random pruning.

**Strengths:**

The differentiator of these paper versus paper on other sketching-based method is the detailed description on how they enable download compression and the back-propagation step, such that the clients never store $d^2$ weight metrics or gradients.

**Weaknesses:**

1. The proposed method is similar to existing sketching-based method, which the authors also cited in the related work. The differentiator of this paper is
1. where the sketching operation happened
2. how to implement them and compute back-propagation efficiently and avoid the $d^2$ memory(section 4).

Hence I believe to making the implementation generic on top of popular deep learning frameworks (e.g PyTorch, TensorFlow) are important.

FetchSGD: Communication-Efficient Federated Learning with Sketching (Rothchild et al., 2020)

**Questions:**

1. the number of clients in i.i.d setting is very small (N=4 and N=10), and the number is unreported for the non-i.i.d setting. Can you report the number of clients in the non-i.i.d setting? How does it work under large number of clients situation?

2. Do you have an open-source implementation of the method? I think making the back-prop for the sketched matrices (equation 5) general and the experiments reproducible can greatly improve the value of the paper.

3. Can you provide more description on how Random and L1 pruning were applied in Figure (4)? L1 pruning is among the best sparsification methods despite of their simplicity (for example Lottery Ticket Hypothesis). In Figure (4) there is a big gap between L1 pruning and the proposed method.

---

### Official Review · Reviewer_p5E1 · 2023-11-06

**Soundness:** 2 fair
**Presentation:** 3 good
**Contribution:** 2 fair
**Rating:** 3
**Confidence:** 3

**Summary:**

This study explores the Federated Learning framework for on-device Machine Learning. In Federated Learning, communication is a crucial limiting factor. Consequently, this research delves into bi-directional compression, wherein both the global model and clients' updates are compressed. The authors introduce a novel algorithm, Comfetch, which employs sketch operators for compression. They also offer convergence guarantees for non-convex scenarios and present experimental results for deep convolutional networks.

**Strengths:**

This paper stands out for its meticulously structured framework, which enhances its overall readability and comprehension.

The abstract serves as a skillfully crafted summary, succinctly encapsulating the fundamental concepts and contributions that are elaborated upon throughout the paper. This succinct summary effectively provides readers with a clear understanding of the paper's scope and objectives.

The introduction, equally impressive in its organization, offers readers a well-defined roadmap for the research. It not only outlines the research problem but also provides a comprehensive rationale, setting the stage for the ensuing exploration.

One notable aspect is the comprehensive treatment of critical components, including Model Transmission and Download, Client Update, and Backwards Pass and Uplink. The meticulous elucidation of these components, along with the derivation of memory complexity, adds depth and clarity to the paper's technical content.

Additionally, the paper excels in presenting its definitions, lemmas, and theorems with precision and clarity. This clarity greatly aids readers in grasping the core mathematical and theoretical foundations on which the research is built.

The experimental section is particularly praiseworthy for its comprehensive and well-structured presentation. The detailed explanation not only outlines the experiments conducted but also provides insight into the methodology, results, and their implications.

In summary, the paper's structural organization, from the abstract to the experimental section, greatly enhances its accessibility and contributes to a better understanding of the research it presents.

**Weaknesses:**

**Introduction**

>This private paradigm has found use in a wide breadth of tasks such as speech prediction, document classification, computer vision, healthcare, and finance.

This sentence lacks proper citations. Please provide multiple citations to support this statement.

>There has been an abundance of progress towards improving communication-efficiency via reduced complexity of outgoing model updates (Konečnỳ et al., 2016 Haddadpour et al., 2021; Ivkin et al. 2019; Rothchild et al., 2020; Reisizadeh et al., 2020; Horvóth et al., 2022; Safaryan et al., 2022. Khirirat et al., 2018).

The reference contains an error; it should be cited as (Horvath et al., 2022).

Horváth, S., Ho, C. Y., Horváth, L. U., Sahu, A. N., Canini, M., & Richtárik, P. (2022). Natural Compression for Distributed Deep Learning. Proceedings of Machine Learning Research vol, 145, 1-40.

I recommend a comprehensive review of all references with meticulous attention to detail. This review should focus on verifying that the references are not only free from typographical errors but also consistent in terms of formatting, citation style, and other relevant bibliographic elements. This attention to detail will ensure that the references are accurate, reliable, and maintain a high standard of quality in your work.

>Our contributions.

The current structure of the contribution section presents information in a single paragraph using sequential numbers. To improve the clarity of the text and make it more reader-friendly, I propose reformatting this section by incorporating bullet points. This would allow for a more organized and visually distinct presentation of each individual contribution, making it easier for readers to understand and appreciate the distinct aspects of the work.

**Relates Work**
In my opinion, this section has several significant issues.

Firstly, in the "Network Compression" subsection, the referenced works are not current, with the most recent citation dating back to 2019. The majority of the other papers mentioned fall within the 2015-2016 timeframe. Given the rapid evolution of the Federated Learning field, it is crucial to provide an up-to-date review. I strongly recommend incorporating more recent papers to address this concern.

>Bi-directional compression works are still uncommon: to the best of our knowledge, (Dorfman et al., 2023), is the only other work to consider compression of model weights while retaining dimensionality, but convergence guarantees are only provided under aggressive assumptions of lossless decompression and experiments are only performed on relatively small models such as ResNet-9.

This statement can be a bit unclear. Although this particular paper cites only one source, it's essential to emphasize that there is a broader collection of papers that delve into the topic of model compression within the context of distributed learning. In other words, the citation provided may be limited, but there is a body of literature that explores the subject of model compression in the context of distributed learning, offering a more comprehensive perspective on the topic:

Caldas, S., Konečny, J., McMahan, H. B., & Talwalkar, A. (2018). Expanding the reach of federated learning by reducing client resource requirements. arXiv preprint arXiv:1812.07210.

Chraibi, S., Khaled, A., Kovalev, D., Richtárik, P., Salim, A., & Takáč, M. (2019). Distributed fixed point methods with compressed iterates. arXiv preprint arXiv:1912.09925.

Shulgin, E., & Richtárik, P. (2022, August). Shifted compression framework: Generalizations and improvements. In Uncertainty in Artificial Intelligence (pp. 1813-1823). PMLR.

Additionally, it's worth noting that there exist several papers that delve into the training of independent subnetworks, and these papers also incorporate model compression mechanisms as part of their methodologies. This signifies that there is a significant body of research dedicated to exploring independent subnetwork training with integrated model compression techniques, expanding the scope of knowledge within this domain:

Yuan, B., Wolfe, C. R., Dun, C., Tang, Y., Kyrillidis, A., & Jermaine, C. (2022). Distributed learning of fully connected neural networks using independent subnet training. Proceedings of the VLDB Endowment, 15(8), 1581-1590.

Shulgin, E., & Richtárik, P. (2023). Towards a better theoretical understanding of independent subnetwork training. arXiv preprint arXiv:2306.16484.

I kindly urge you to give these papers thorough consideration and conduct a comprehensive comparison of their findings. Incorporating the results and insights derived from such a comparative analysis into your review would greatly enhance its depth and value. By examining these papers in detail and contrasting their outcomes, you can offer readers a more robust and insightful review of the subject matter.

>Other bi-directional works transmit compressed gradients (Philippenko & Dieuleveut, 2020; Tang et al., 2019; Gruntkowska et al., 2023; Zheng et al., 2019) and require restoration inside local memory which is equivalent to storing a fullysized architecture, which Comfetch avoids.

This statement appears to be inaccurate and unclear. Firstly, it lacks clarity regarding the concept of "restoration inside local memory" and why it is suggested that employing such restoration is equivalent to storing a full-sized model. It is crucial to provide a detailed explanation of this concept to eliminate any confusion.

From my understanding, if the compression operator is a Rand-K or Top-K compressor, only $K$ components out of $d$ components are non-zero. The application of such compressors results in the utilization of submodels instead of retaining the entire full-sized model. This differs from the assertion that it is equivalent to storing a full-sized model. To address this issue, please clarify the intended meaning of your claim or adjust it accordingly. As it stands, this claim is inappropriate and likely to cause confusion.

**Model Update**
>The error-feedback term et allows for the correction of error associated with our gradient approximations $g^l$.

The present study employs an error feedback mechanism for compression schemes. Nevertheless, it is worth noting that the work in question does not incorporate an assessment or comparative analysis with the latest iteration of error feedback mechanisms, specifically EF-21. This omission highlights an opportunity to further enhance the study by examining and contrasting the results achieved with the more recent EF-21 approach. Such a comparative analysis would contribute to a more comprehensive and up-to-date understanding of the subject matter.

Richtárik, P., Sokolov, I., & Fatkhullin, I. (2021). EF21: A new, simpler, theoretically better, and practically faster error feedback. Advances in Neural Information Processing Systems, 34, 4384-4396.

Fatkhullin, I., Sokolov, I., Gorbunov, E., Li, Z., & Richtárik, P. (2021). EF21 with bells & whistles: Practical algorithmic extensions of modern error feedback. arXiv preprint arXiv:2110.03294.

Fatkhullin, I., Tyurin, A., & Richtárik, P. (2023). Momentum Provably Improves Error Feedback!. arXiv preprint arXiv:2305.15155.

Could you kindly provide a thorough evaluation and comparison with the recently developed method? This would greatly enrich the analysis and help to highlight the advantages and limitations of the new approach in relation to the subject matter under investigation.

>Assumption 2 (Unbiased and Bounded). All stochastic gradients $g$ of $f(w)$ are unbiased and bounded,
$$
\mathbb{E} g=\nabla f(w) \text { and } \mathbb{E}\Vert g\Vert^2 \leq G^2 .
$$

This assumption implies that both the stochastic gradient and the true gradient are bounded because the second moment is bounded, and the stochastic gradient is unbiased:

$$ \mathbb{E}\Vert g\Vert^2 = \mathbb{E}\Vert g - \mathbb{E}g \Vert^2 + \Vert \mathbb{E}g \Vert^2  = Var(g) + \Vert \mathbb{E}g \Vert^2 = Var(g) + \Vert \nabla f(x) \Vert^2\leq G^2.$$

Since $Var(g)\geq 0$, $\Vert \nabla f(x) \Vert^2\geq 0$ and $ Var(g) + \Vert \nabla f(x) \Vert^2\leq G^2$ we have that $Var(g)$ and $\Vert \nabla f(x) \Vert^2$ are bounded.

Given that $\Vert \nabla f(x) \Vert^2 \leq C_1^2$ and $C_1^2\geq 0$, it follows that $\Vert \nabla f(x) \Vert \leq C_1$. This implies a constraint on the loss function, signifying its limited variation. Even a relatively simple quadratic function, like $$f(x) = \Vert Ax-b \Vert^2,$$ fails to meet the requirement of a bounded gradient assumption.

A more suitable assumption would be the bounded variance assumption, $Var(g) \leq \sigma^2,$ which is commonly employed in many contemporary Federated Learning papers, as seen in numerous examples such as

Karimireddy, S. P., Kale, S., Mohri, M., Reddi, S., Stich, S., & Suresh, A. T. (2020, November). Scaffold: Stochastic controlled averaging for federated learning. In International conference on machine learning (pp. 5132-5143). PMLR.

Patel, K. K., Wang, L., Woodworth, B. E., Bullins, B., & Srebro, N. (2022). Towards optimal communication complexity in distributed non-convex optimization. Advances in Neural Information Processing Systems, 35, 13316-13328.

I've reviewed the convergence proof, and with all due respect, it does not present a significant theoretical challenge or an interesting aspect. The core of this proof rests upon the strong assumption of having both bounded and unbiased stochastic gradients. This assumption, while simplifying the mathematical analysis, essentially paves the way for a straightforward proof that builds upon previously established lemmas. Consequently, the proof may not offer the same level of theoretical insight or intrigue as proofs that tackle more intricate and nuanced scenarios.

Furthermore, it's important to note that this assumption is certainly not applicable to Deep Neural Networks in a general sense, and specifically, it cannot be extended to Residual Networks (Res-Nets). The complexities and intricacies of deep neural networks, especially Res-Net architectures, introduce a range of challenges and behaviors that deviate from the simplicity of the assumptions made in the current context. These networks involve non-linear activations, skip connections, and deep layer structures that significantly impact the behavior of gradients, making it inappropriate to apply the same assumption regarding bounded and unbiased stochastic gradients. As a result, the practicality and relevance of this assumption in the realm of deep learning, especially for Res-Net models, are quite limited.

>Assumption 3 is a variant on a common heavy-hitter assumption suggested in the convergence theorem of Fet chSGD to ensure successful error-feedback (Rothchild et al., 2020). Heavy hitters are also required in the convergence analysis of Sket ched-SGD (Ivkin et al., 2019). In this version, we are requiring $\nabla_w \mathcal{L}\left(H^{\top} H w\right)$ to contain a heavy hitter.

Could you kindly provide further elaboration on this assumption? What insights does it offer, and why is it deemed necessary? It would be greatly appreciated if you could offer illustrative examples to demonstrate how this assumption is applied in this context.

Additionally, it would be advantageous to include a table that provides a theoretical comparison between the novel findings and the results obtained in previous studies. This would offer a structured and concise means of highlighting the distinctions and contributions of the new results in relation to the existing body of research. Such a table can serve as a valuable reference point for readers seeking to understand the theoretical advancements presented in the study and their significance in the broader context of prior work.

**Experiments**

There are several issues with the experimental results. Firstly, the plots in Figures 2, 3, and 4 are relatively small, which makes it challenging to discern all the details within them. Furthermore, these plots lack markers, rendering them inaccessible to individuals with color blindness. Therefore, I kindly request that you consider adjusting the plots to address these concerns.

Expanding on this, increasing the size of the figures and incorporating distinctive markers or patterns would greatly improve the readability and accessibility of the plots, ensuring that they are more comprehensible and inclusive for all readers, regardless of their visual abilities.

A significant concern that arises within the experimental section pertains to the comparative analysis of the new method. In this section, the new approach is exclusively pitted against various degrees of compression and the uncompressed version, which is typically represented by FedAvg. While this internal comparison provides valuable insights into the performance of the new method under different levels of compression, it falls short in one crucial aspect.

Notably, the experimental section neglects to include a comparison with other bi-directional methods that were introduced and discussed in the introductory sections of the paper. This omission is noteworthy as it limits the scope and depth of the experimentation. Readers are left with a gap in their understanding, as they are not provided with insights into how the new method stacks up against its contemporaries or other relevant techniques in the field.

By failing to incorporate these comparative analyses with other bi-directional methods, the experimental section misses an opportunity to offer a more comprehensive evaluation of the new method's strengths, weaknesses, and overall effectiveness. In addressing this concern, the experimental section could be enhanced to provide a more well-rounded and informative assessment of the new method's performance in relation to existing alternatives.

**Questions:**

Please check questions raised in Weaknesses section.

Furthermore, I would like to propose the consideration of non-convex analysis, specifically incorporating the Polyak-Lojasiewicz condition. This addition would expand the scope of the analysis and provide a more comprehensive exploration of the problem.